

COMPUTO

ISSN 2824-7795

# Peerannot: classification for crowd-sourced image datasets with Python

Tanguy Lefort [1]    IMAG, Univ Montpellier, CNRS, Inria, LIRMM
Benjamin Charlier    IMAG, Univ Montpellier, CNRS
Alexis Joly    Inria, LIRMM, Univ Montpellier, CNRS
Joseph Salmon    IMAG, Univ Montpellier, CNRS, IUF

Date published: 2024-01-29    Last modified: 2024-01-29

## Abstract

Crowdsourcing is a quick and easy way to collect labels for large datasets, involving many workers. However, workers often disagree with each other. Sources of error can arise from the workers' skills, but also from the intrinsic difficulty of the task. We present `peerannot`: a Python library for managing and learning from crowdsourced labels for classification. Our library allows users to aggregate labels from common noise models or train a deep learning-based classifier directly from crowdsourced labels. In addition, we provide an identification module to easily explore the task difficulty of datasets and worker capabilities.

*Keywords:* crowdsourcing, label noise, task difficulty, worker ability, classification

# Contents

---

[1]Corresponding author: tanguy.lefort@umontpellier.fr

# 1   Introduction: crowdsourcing in image classification

Image datasets widely use crowdsourcing to collect labels, involving many workers who can annotate images for a small cost (or even free for instance in citizen science) and faster than using expert labeling. Many classical datasets considered in machine learning have been created with human intervention to create labels, such as CIFAR-10, (Krizhevsky and Hinton 2009), ImageNet (Deng et al. 2009) or Pl@ntnet (Garcin et al. 2021) in image classification, but also COCO (Lin et al. 2014), solar photovoltaic arrays (Kasmi et al. 2023) or even macro litter (Chagneux et al. 2023) in image segmentation and object counting.

Crowdsourced datasets induce at least three major challenges to which we contribute with `peerannot`:

1) **How to aggregate multiple labels into a single label from crowdsourced tasks?** This occurs for example when dealing with a single dataset that has been labeled by multiple workers with disagreements. This is also encountered with other scoring issues such as polls, reviews, peer-grading, *etc.* In our framework this is treated with the `aggregate` command, which given multiple labels, infers a label. From aggregated labels, a classifier can then be trained using the `train` command.

2) **How to learn a classifier from crowdsourced datasets?** Where the first question is bound by aggregating multiple labels into a single one, this considers the case where we do not need a single label to train on, but instead train a classifier on the crowdsourced data, with the motivation to perform well on a testing set. This end-to-end vision is common in machine learning, however, it requires the actual tasks (the images, texts, videos, *etc.*) to train on – and in crowdsourced datasets publicly available, they are not always available. This is treated with the `aggregate-deep` command that runs strategies where the aggregation has been transformed into a deep learning optimization problem.

3) **How to identify good workers in the crowd and difficult tasks?** When multiple answers are given to a single task, looking for who to trust for which type of task becomes necessary

to estimate the labels or later train a model with as few noise sources as possible. The module `identify` uses different scoring metrics to create a worker and/or task evaluation. This is particularly relevant considering the gamification of crowdsourcing experiments (Servajean et al. 2016)

The library `peerannot` addresses these practical questions within a reproducible setting. Indeed, the complexity of experiments often leads to a lack of transparency and reproducible results for simulations and real datasets. We propose standard simulation settings with explicit implementation parameters that can be shared. For real datasets, `peerannot` is compatible with standard neural network architectures from the `Torchvision` (Marcel and Rodriguez 2010) library and `Pytorch` (Paszke et al. 2019), allowing a flexible framework with easy-to-share scripts to reproduce experiments.

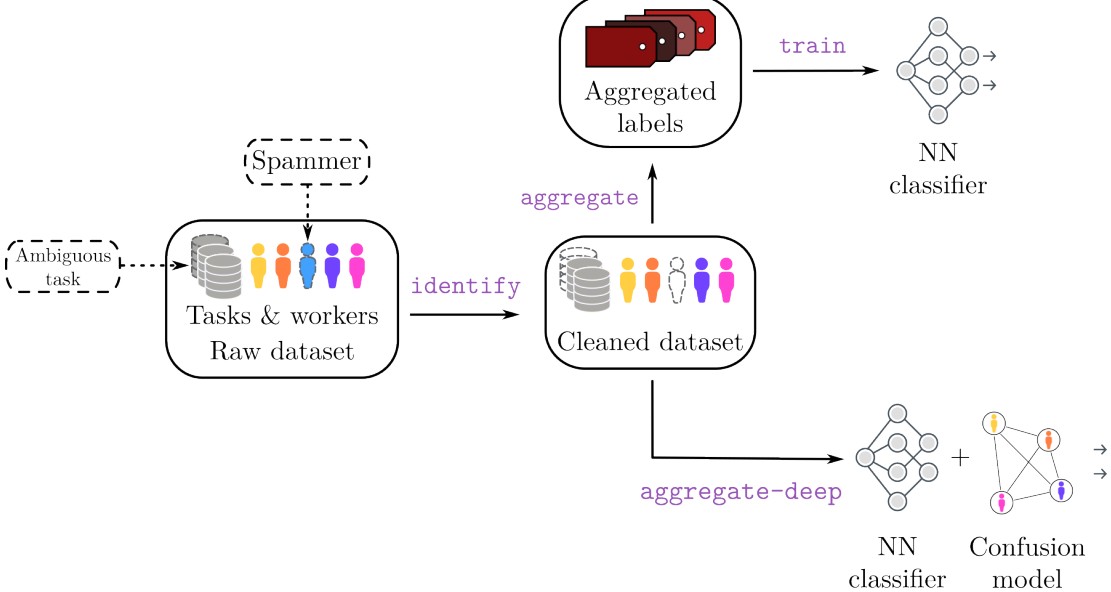

Figure 1: From crowdsourced labels to training a classifier neural network, the learning pipeline using the `peerannot` library. An optional preprocessing step using the `identify` command allows us to remove the worst-performing workers or images that can not be classified correctly (very bad quality for example). Then, from the cleaned dataset, the `aggregate` command may generate a single label per task from a prescribed strategy. From the aggregated labels we can train a neural network classifier with the `train` command. Otherwise, we can directly train a neural network classifier that takes into account the crowdsourcing setting in its architecture using `aggregate-deep`.

## 2 Notation and package structure

### 2.1 Crowdsourcing notation

Let us consider the classical supervised learning classification framework. A training set $\mathscr{D} = \{(x_i, y_i^\star)\}_{i=1}^{n_{\text{task}}}$ is composed of $n_{\text{task}}$ tasks $x_i \in \mathscr{X}$ (the feature space) with (unknown) true label $y_i^\star \in [K] = \{1, \ldots, K\}$ one of the $K$ possible classes. In the following, the tasks considered are generally RGB images. We use the notation $\sigma(\cdot)$ for the softmax function. In particular, given a classifier $\mathscr{C}$ with logits outputs, $\sigma(\mathscr{C}(x_i))_{[1]}$ represents the largest probability and we can sort the probabilities as $\sigma(\mathscr{C}(x_i))_{[1]} \geq \sigma(\mathscr{C}(x_i))_{[2]} \geq \cdots \geq \sigma(\mathscr{C}(x_i))_{[K]}$. The indicator function is denoted $\mathbf{1}(\cdot)$. We use the $i$ index notation to range over the different tasks and the $j$ index notation for the workers in the crowdsourcing experiment. Note that indices start at position 1 in the equation to follow mathematical standard notation but it should be noted that, as this is a `Python` library, in the code

indices start at the 0 position.

With crowdsourced data the true label of a task $x_i$, denoted $y_i^\star$ is unknown, and there is no single label that can be trusted as in standard supervised learning (even on the train set!). Instead, there is a crowd of $n_{\text{worker}}$ workers from which multiple workers $(w_j)_j$ propose a label $(y_i^{(j)})_j$. These proposed labels are used to estimate the true label. The set of workers answering the task $x_i$ is denoted by

$$\mathscr{A}(x_i) = \left\{ j \in [n_{\text{worker}}] \ : \ w_j \text{ answered } x_i \right\}. \tag{1}$$

The cardinal $|\mathscr{A}(x_i)|$ is called the feedback effort on the task $x_i$. Note that the feedback effort can not exceed the total number of workers $n_{\text{worker}}$. Similarly, one can adopt a worker point of view: the set of tasks answered by a worker $w_j$ is denoted

$$\mathscr{T}(w_j) = \left\{ i \in [n_{\text{task}}] \ : \ w_j \text{ answered } x_i \right\}. \tag{2}$$

The cardinal $|\mathscr{T}(w_j)|$ is called the workload of $w_j$. The final dataset can then be decomposed as:

$$\mathscr{D}_{\text{train}} := \bigcup_{i \in [n_{\text{task}}]} \left\{ (x_i, (y_i^{(j)})) \text{ for } j \in \mathscr{A}(x_i) \right\} = \bigcup_{j \in [n_{\text{worker}}]} \left\{ (x_i, (y_i^{(j)})) \text{ for } i \in \mathscr{T}(w_j) \right\} \ .$$

In this article, we do not address the setting where workers report their self-confidence (Yasmin et al. 2022), nor settings where workers are presented a trapping set – *i.e.,* a subset of tasks where the true label is known to evaluate them with known labels (Khattak 2017).

## 2.2 Storing crowdsourced datasets in `peerannot`

Crowdsourced datasets come in various forms. To store crowdsourcing datasets efficiently and in a standardized way, `peerannot` proposes the following structure, where each dataset corresponds to a folder. Let us set up a toy dataset example to understand the data structure and how to store it.

---
**Listing 1** Dataset storage tree structure.

```
datasetname
        train
                ...
                images
                ...
        val
        test
        metadata.json
        answers.json
```
---

The `answers.json` file stores the different votes for each task as described in Figure 2. This `.json` is the rosetta stone between the task ids and the images. It contains the tasks' id, the workers's id and the proposed label for each given vote. Furthermore, storing labels in a dictionary is more memory-friendly than having an array of size (`n_task,n_worker`) and writing $y_i^{(j)} = -1$ when the worker $w_j$ did not see the task $x_i$ and $y_i^{(j)} \in [K]$ otherwise.

In Figure 2, there are three tasks, $n_{\text{worker}} = 4$ workers and $K = 2$ classes. Any available task should be stored in a single file whose name follows the convention described in Listing 1. These files are spread into a `train`, `val` and `test` subdirectories as in `ImageFolder` datasets from `torchvision`

Finally, a `metadata.json` file includes relevant information related to the crowdsourcing experiment such as the number of workers, the number of tasks, *etc.* For example, a minimal `metadata.json` file for the toy dataset presented in Figure 2 is:

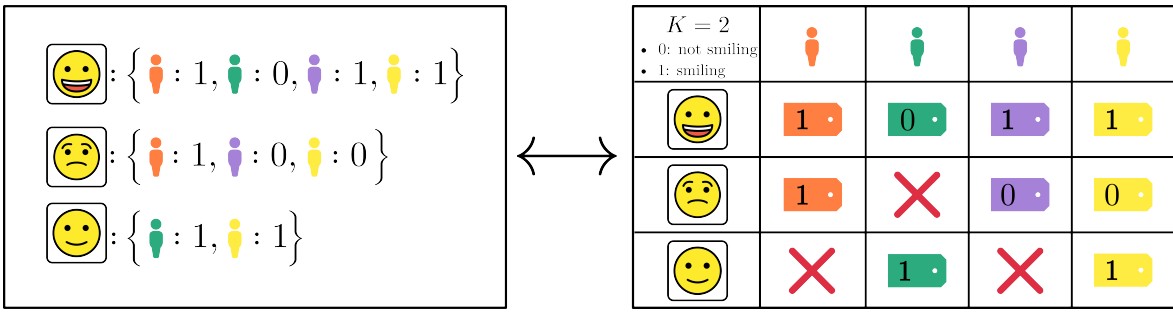

Figure 2: Data storage for the `toy-data` crowdsourced dataset, a binary classification problem ($K = 2$, smiling/not smiling) on recognizing smiling faces. (left: how data is stored in `peerannot` in a file `answers.json`, right: data collected)

```
{
    "name": "toy-data",
    "n_classes": 2,
    "n_workers": 4,
    "n_tasks": 3
}
```

The `toy-data` example dataset is available as an example in the peerannot repository. Classical datasets in crowdsourcing such as `CIFAR-10H` (Peterson et al. 2019) and `LabelMe` (Rodrigues, Pereira, and Ribeiro 2014) can be installed directly using `peerannot`. To install them, run the `install` command from `peerannot`:

For both `CIFAR-10H` and `LabelMe`, the dataset was was originally released for standard supervised learning (classification). Both datasets has been reannotated by a crowd or workers. These labels are used as true labels in evaluations and visualizations. Examples of `CIFAR-10H` images are available in Figure 16, and `LabelMe` examples in Figure 17 in Appendix. Crowdsourcing votes, however, bring information about possible confusions (see Figure 3 for an example with `CIFAR-10H` and Figure 4 with `LabelMe`).

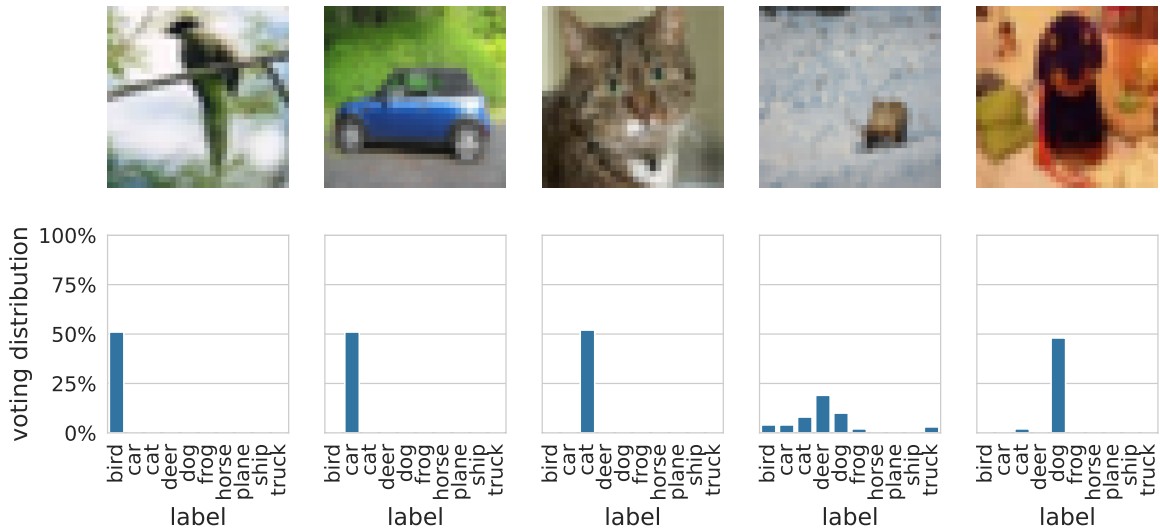

Figure 3: Example of crowdsourced images from CIFAR-10H. Each task has been labeled by multiple workers. We display the associated voting distribution over the possible classes.

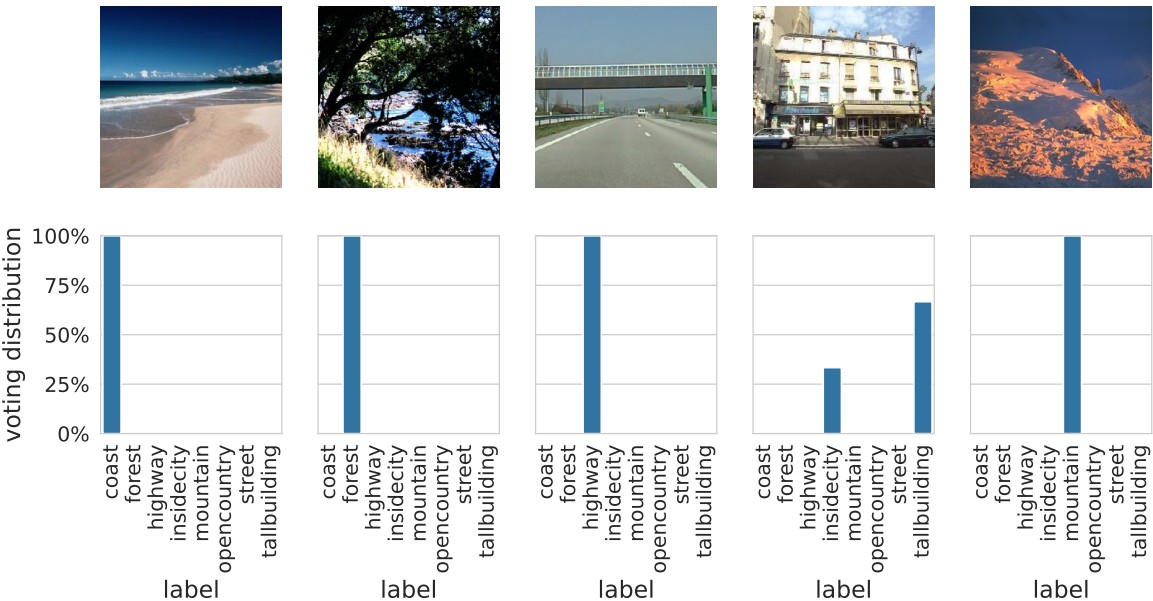

Figure 4: Example of crowdsourced images from LabelMe. Each task has been labeled by multiple workers. We display the associated voting distribution over the possible classes.

# 3 Aggregation strategies in crowdsourcing

The first question we address with peerannot is: *How to aggregate multiple labels into a single label from crowdsourced tasks?* The aggregation step can lead to two types of learnable labels $\hat{y}_i \in \Delta_K$ (where $\Delta_K$ is the simplex of dimension $K - 1$: $\Delta_K = \{p \in \mathbb{R}^K : \sum_{k=1}^K p_k = 1, p_k \geq 0\}$) depending on the use case for each task $x_i$, $i = 1, \dots, n_{\text{task}}$:

- a **hard** label: $\hat{y}_i$ is a Dirac distribution, this can be encoded as a classical label in $[K]$,
- a **soft** label: $\hat{y}_i \in \Delta_K$ can represent any probability distribution on $[K]$. In that case, each coordinate of the $K$-dimensional vector $\hat{y}_i$ represents the probability of belonging to the given class.

Learning from soft labels has been shown to improve learning performance and make the classifier learn the task ambiguity (Zhang et al. 2018; Peterson et al. 2019; Park and Caragea 2022). However, crowdsourcing is often used as a stepping stone to create a new dataset. We usually expect a classification dataset to associate a task $x_i$ to a single label and not a full probability distribution. In this case, we recommend releasing the anonymous answered labels and the aggregation strategy used to reach a consensus on a single label. With peerannot, both soft and hard labels can be produced.

Note that when a strategy produces a soft label, a hard label can be easily induced by taking the mode, *i.e.,* the class achieving the maximum probability.

## 3.1 Classical models

We list below the most classical aggregation strategies used in crowdsourcing.

### 3.1.1 Majority vote (MV)

The most intuitive way to create a label from multiple answers for any type of crowdsourced task is to take the majority vote (MV). Yet, this strategy has many shortcomings (James 1998) – there is no

noise model, no worker reliability estimated, no task difficulty involved and especially no way to remove poorly performing workers. This standard choice can be expressed as:

$$\hat{y}_i^{\mathrm{MV}} = \underset{k \in [K]}{\arg\max} \sum_{j \in \mathscr{A}(x_i)} \mathbf{1}_{\{y_i^{(j)}=k\}} \ .$$

### 3.1.2 Naive soft (NS)

One pitfall with MV is that the label produced is hard, hence the ambiguity is discarded by construction. A simple remedy consists in using the Naive Soft (NS) labeling, *i.e.,* output the empirical distribution as the task label:

$$\hat{y}_i^{\mathrm{NS}} = \left( \frac{1}{|\mathscr{A}(x_i)|} \sum_{j \in \mathscr{A}(x_i)} \mathbf{1}_{\{y_i^{(j)}=k\}} \right)_{j \in [K]} \ .$$

With the NS label, we keep the ambiguity, but all workers and all tasks are put on the same level. In practice, it is known that each worker comes with their abilities, thus modeling this knowledge can produce better results.

### 3.1.3 Dawid and Skene (DS)

Refining the aggregation, researchers have proposed a noise model to take into account the workers' abilities. The Dawid and Skene's (DS) model (Dawid and Skene 1979) is one of the most studied (Gao and Zhou 2013) and applied (Servajean et al. 2017; Rodrigues and Pereira 2018). These types of models are most often optimized using EM-based procedures. Assuming the workers are answering tasks independently, this model boils down to model pairwise confusions between each possible class. Each worker $w_j$ is assigned a confusion matrix $\pi^{(j)} \in \mathbb{R}^{K \times K}$ as described in Section 3. The model assumes that for a task $x_i$, conditionally on the true label $y_i^\star = k$ the label distribution of the worker's answer follows a multinomial distribution with probabilities $\pi_{k,\cdot}^{(j)}$ for each worker. Each class has a prevalence $\rho_k = \mathbb{P}(y_i^\star = k)$ to appear in the dataset. Using the independence between workers, we obtain the following likelihood to maximize, with latent variables $\rho, \pi = \{\pi^{(j)}\}_j$ and unobserved variables $(y_i^{(j)})_{i,j}$:

$$\arg\max_{\rho,\pi} \prod_{i \in [n_{\mathrm{task}}]} \prod_{k \in [K]} \left[ \rho_k \prod_{j \in [n_{\mathrm{worker}}]} \prod_{\ell \in [K]} \left( \pi_{k,\ell}^{(j)} \right)^{\mathbf{1}_{\{y_i^{(j)}=\ell\}}} \right].$$

When the true labels are not available, the data comes from a mixture of categorical distributions. To retrieve ground truth labels and be able to estimate these parameters, Dawid and Skene (1979) have proposed to consider the true labels as additional unknown parameters. In this case, denoting $T_{i,k} = \mathbf{1}_{\{y_i^\star = k\}}$ the vectors of label class indicators for each task, the likelihood with known true labels is:

$$\arg\max_{\rho,\pi,T} \prod_{i \in [n_{\mathrm{task}}]} \prod_{k \in [K]} \left[ \rho_k \prod_{j \in [n_{\mathrm{worker}}]} \prod_{\ell \in [K]} \left( \pi_{k,\ell}^{(j)} \right)^{\mathbf{1}_{\{y_i^{(j)}=\ell\}}} \right]^{T_{i,k}} .$$

This framework allows to estimate $\rho, \pi, T$ with an EM algorithm as follows:

- With the MV strategy, get an initial estimate of the true labels $T$.
- Estimate $\rho$ and $\pi$ knowing $T$ using maximum likelihood estimators.
- Update $T$ knowing $\rho$ and $\pi$ using Bayes formula.
- Repeat until convergence of the likelihood.

The final aggregated soft labels are $\hat{y}_i^{\text{DS}} = T_{i,\cdot}$. Note that DS also provides the estimated confusion matrices $\hat{\pi}^{(j)}$ for each worker $w_j$.

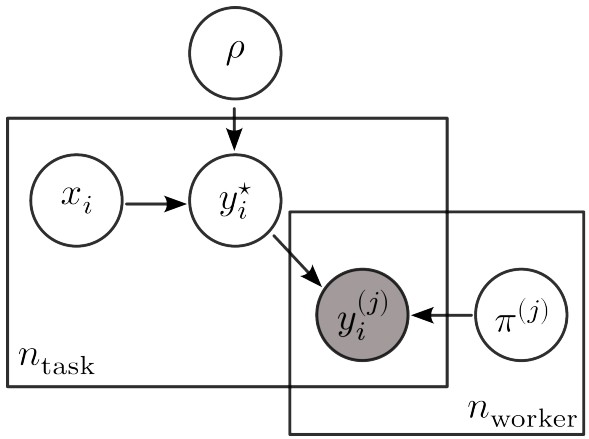

Figure 5: Bayesian plate notation for the DS model

- $\rho \in \Delta_K$: prevalence
- $x_i \in \mathcal{X}$: task
- $y_i^{\star} \in [K]$: true label (unobserved)
- $y_i^{(j)} \in [K]$: label observed
- $\pi^{(j)} \in \mathbb{R}^{K \times K}$: confusion matrix

### 3.1.4 Variations around the DS model

Many variants of the DS model have been proposed in the literature, using Dirichlet priors on the confusion matrices (Passonneau and Carpenter 2014), using $1 \leq L \leq n_{\text{worker}}$ clusters of workers (Imamura, Sato, and Sugiyama 2018) (DSWC) or even faster implementation that produces only hard labels (Sinha, Rao, and Balasubramanian 2018).

In particular, the DSWC strategy (Dawid and Skene with Worker Clustering) highly reduces the dimension of the parameters in the DS model. In the original model, there are $K^2 \times n_{\text{worker}}$ parameters to be estimated for the confusion matrices only. The DSWC model reduces them to $K^2 \times L + L$ parameters. Indeed, there are $L$ confusion matrices $\Lambda = \{\Lambda_1, \dots, \Lambda_L\}$ and the confusion matrix of a cluster is assumed drawn from a multinomial distribution with weights $(\tau_1, \dots, \tau_L) \in \Delta_L$ over $\Lambda$, such that $\mathbb{P}(\pi^{(j)} = \Lambda_\ell) = \tau_\ell$.

### 3.1.5 Generative model of Labels, Abilities, and Difficulties (GLAD)

Finally, we present the GLAD model (Whitehill et al. 2009) that not only takes into account the worker's ability, but also the task difficulty in the noise model. The likelihood is optimized using an EM algorithm to recover the soft label $\hat{y}_i^{\text{GLAD}}$.

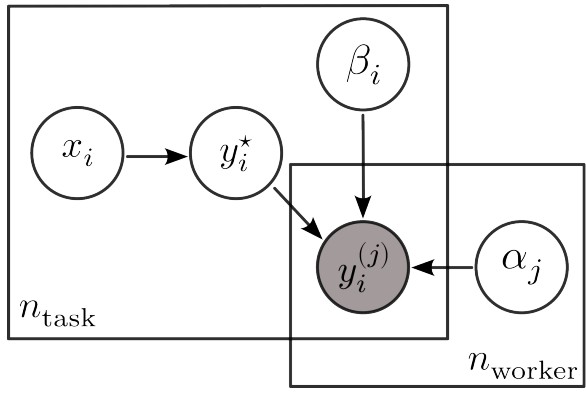

Figure 6: Bayesian plate notation for the GLAD model

- $x_i \in \mathcal{X}$: task
- $y_i^{\star} \in [K]$: true label (unobserved)
- $y_i^{(j)} \in [K]$: label observed
- $\alpha_j \in \mathbb{R}$: worker reliability
- $\beta_i \in \mathbb{R}_{\star}^+$: task difficulty

Denoting $\alpha_j \in \mathbb{R}$ the worker ability (the higher the better) and $\beta_i \in \mathbb{R}_\star^+$ the task's difficulty (the higher the easier), the model noise is:

$$\mathbb{P}(y_i^{(j)} = y_i^\star | \alpha_j, \beta_i) = \frac{1}{1 + \exp(-\alpha_j \beta_i)} \quad .$$

GLAD's model also assumes that the errors are uniform across wrong labels, thus:

$$\forall k \in [K], \ \mathbb{P}(y_i^{(j)} = k | y_i^\star \neq k, \alpha_j, \beta_i) = \frac{1}{K-1}\left(1 - \frac{1}{1 + \exp(-\alpha_j \beta_i)}\right) \quad .$$

This results in estimating $n_{\text{worker}} + n_{\text{task}}$ parameters.

### 3.1.6 Aggregation strategies in `peerannot`

All of these aggregation strategies – and more – are available in the `peerannot` library from the `peerannot.models` module. Each model is a class object in its own `Python` file. It inherits from the `CrowdModel` template class and is defined with at least two methods:

- `run`: includes the optimization procedure to obtain needed weights (*e.g.,* the EM algorithm for the DS model),
- `get_probas`: returns the soft labels output for each task.

## 3.2 Experiments and evaluation of label aggregation strategies

One way to evaluate the label aggregation strategies is to measure their accuracy. This means that the underlying ground truth must be known – at least for a representative subset. This is the case in simulation settings where the ground truth is available. As the set of $n_{\text{task}}$ can be seen as a training set for a future classifier, we denote this metric AccTrain on a dataset $\mathscr{D}$ for some given aggregated label $(\hat{y}_i)_i$ as:

$$\text{AccTrain}(\mathscr{D}) = \frac{1}{|\mathscr{D}|} \sum_{i=1}^{|\mathscr{D}|} \mathbf{1}_{\{y_i^\star = \text{argmax}_{k \in [K]}(\hat{y}_i)_k\}} \quad .$$

In the following, we write AccTrain for AccTrain($\mathscr{D}_{\text{train}}$) as we only consider the full training set so there is no ambiguity. The AccTrain computes the number of correctly predicted labels by the aggregation strategy knowing a ground truth. While this metric is useful, in practice there are a few arguable issues:

- the AccTrain metric does not consider the ambiguity of the soft label, only the most probable class, whereas in some contexts ambiguity can be informative,
- in supervised learning one objective is to identify difficult or mislabeled tasks (Pleiss et al. 2020; Lefort et al. 2022), pruning those tasks can easily artificially improve the AccTrain, but there is no guarantee over the predictive performance of a model based on the newly pruned dataset,
- in practice, true labels are unknown, thus this metric would not be computable.

We first consider classical simulation settings in the literature that can easily be created and reproduced using `peerannot`. For each dataset, we present the distribution of the number of workers per task $(|\mathscr{A}(x_i)|)_{i=1,\dots,n_{\text{task}}}$ Equation 1 on the right and the distribution of the number of tasks per worker $(|\mathscr{T}(w_j)|)_{j=1,\dots,n_{\text{worker}}}$ Equation 2 on the left.

### 3.2.1 Simulated independent mistakes

The independent mistakes setting considers that each worker $w_j$ answers follows a multinomial distribution with weights given at the row $y_i^\star$ of their confusion matrix $\pi^{(j)} \in \mathbb{R}^{K \times K}$. Each confusion row in the confusion matrix is generated uniformly in the simplex. Then, we make the matrix diagonally dominant (to represent non-adversarial workers) by switching the diagonal term with the maximum value by row. Answers are independent of one another as each matrix is generated independently and each worker answers independently of other workers. In this setting, the DS model is expected to perform better with enough data as we are simulating data from its assumed noise model.

We simulate $n_{\text{task}} = 200$ tasks and $n_{\text{worker}} = 30$ workers with $K = 5$ possible classes. Each task $x_i$ receives $|\mathscr{A}(x_i)| = 10$ labels. With 200 tasks and 30 workers, asking for 10 leads to around $\frac{200 \times 10}{30} \simeq 67$ tasks per worker (with variations due to randomness in the affectations).

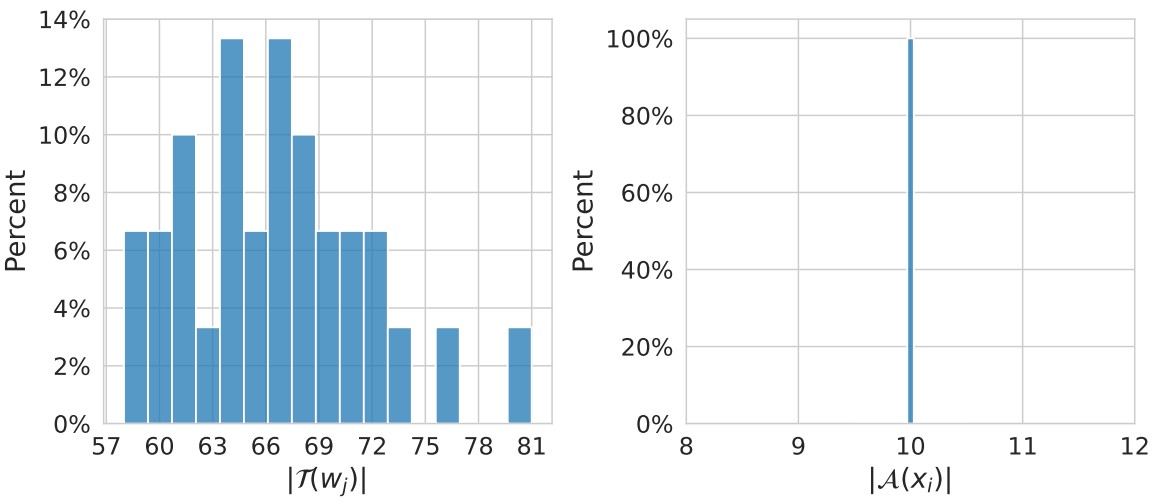

Figure 7: Distribution of number of tasks given per worker (left) and number of labels per task (right) in the independent mistakes setting.

With the obtained answers, we can look at the aforementioned aggregation strategies performance. The `peerannot aggregate` command takes as input the path to the data folder and the aggregation `--strategy/-s` to perform. Other arguments are available and described in the `--help` description.

Table 1: AccTrain metric on simulated independent mistakes considering classical feature-blind label aggregation strategies

Table 1

|  | MV | GLAD | DS | DSWC[L=5] | DSWC[L=10] | NS |
|---|---|---|---|---|---|---|
| AccTrain | 0.760 | 0.775 | 0.890 | 0.775 | 0.770 | 0.760 |

As expected by the simulation framework, Table 1 fits the DS model, thus leading to better accuracy in retrieving the simulated labels for the DS strategy. The MV and NS aggregations do not consider any worker-ability scoring or the task's difficulty and perform the worst.

**Remark:** `peerannot` can also simulate datasets with an imbalanced number of votes chosen uniformly at random between 1 and the number of workers available. For example:

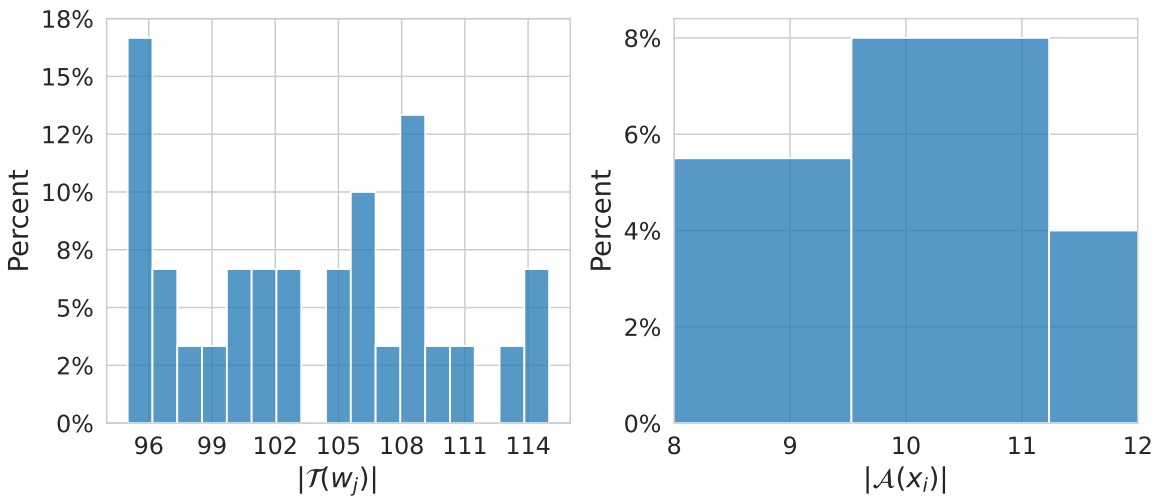

Figure 8: Distribution of the number of tasks given per worker (left) and of the number of labels per task (right) in the independent mistakes setting with voting imbalance enabled.

With the obtained answers, we can look at the aforementioned aggregation strategies performance:

Table 2: AccTrain metric on simulated independent mistakes with an imbalanced number of votes per task considering classical feature-blind label aggregation strategies

Table 2

|          | MV    | GLAD  | DS    | DSWC[L=5] | DSWC[L=10] | NS    |
|----------|-------|-------|-------|-----------|------------|-------|
| AccTrain | 0.825 | 0.810 | 0.895 | 0.845     | 0.840      | 0.830 |

While more realistic, working with an imbalanced number of votes per task can lead to disrupting orders of performance for some strategies (here GLAD is outperformed by other strategies).

### 3.2.2 Simulated correlated mistakes

The correlated mistakes are also known as the student-teacher or junior-expert setting (Cao et al. (2019)). Consider that the crowd of workers is divided into two categories: teachers and students (with $n_{\text{teacher}} + n_{\text{student}} = n_{\text{worker}}$). Each student is randomly assigned to one teacher at the beginning of the experiment. We generate the (diagonally dominant as in Section 3.2.1) confusion matrices of each teacher and the students share the same confusion matrix as their associated teacher. Hence, clustering strategies are expected to perform best in this context. Then, they all answer independently, following a multinomial distribution with weights given at the row $y_i^\star$ of their confusion matrix $\pi^{(j)} \in \mathbb{R}^{K \times K}$.

We simulate $n_{\text{task}} = 200$ tasks and $n_{\text{worker}} = 30$ with 80% of students in the crowd. There are $K = 5$ possible classes. Each task receives $|\mathcal{A}(x_i)| = 10$ labels.

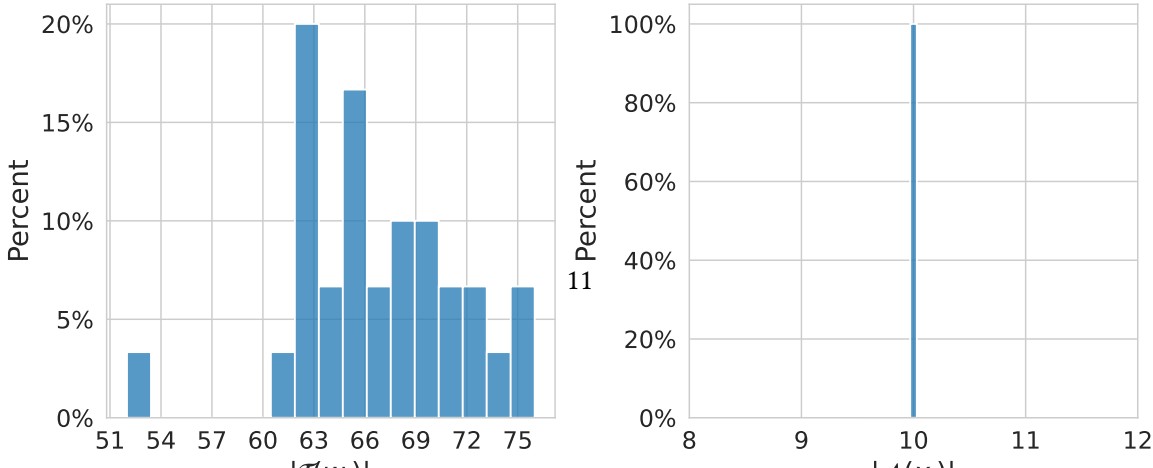

Table 3: AccTrain metric on simulated correlated mistakes considering classical feature-blind label aggregation strategies

| | MV | GLAD | DS | DSWC[L=5] | DSWC[L=6] | DSWC[L=10] | NS |
|---|---|---|---|---|---|---|---|
| AccTrain | 0.705 | 0.645 | 0.755 | 0.795 | 0.780 | 0.815 | 0.690 |

With Table 3, we see that with correlated data (24 students and 6 teachers), using 5 confusion matrices with DSWC[L=5] outperforms the vanilla DS strategy that does not consider the correlations. The best-performing method here estimates only 10 confusion matrices (instead of 30 for the vanilla DS model).

To summarize our simulations, we see that depending on workers answering strategies, different latent variable models perform best. However, these are unknown outside of a simulation framework, thus if we want to obtain labels from multiple responses, we need to investigate multiple models. This can be done easily with `peerannot` as we demonstrated using the `aggregate` module. However, one might not want to generate a label, simply learn a classifier to predict labels on unseen data. This leads us to another module part of `peerannot`.

### 3.3 More on confusion matrices in simulation settings

Moreover, the concept of confusion matrices has been commonly used to represent worker abilities. Let us remind that a confusion matrix $\pi^{(j)} \in \mathbb{R}^{K \times K}$ of a worker $w_j$ is defined such that $\pi_{k,\ell}^{(j)} = \mathbb{P}(y_i^{(j)} = \ell | y_i^\star = k)$. These quantities need to be estimated since no true label is available in a crowd-sourced scenario. In practice, the confusion matrices of each worker is estimated via an aggregation strategy like Dawid and Skene's (Dawid and Skene 1979) presented in Section 3.1.

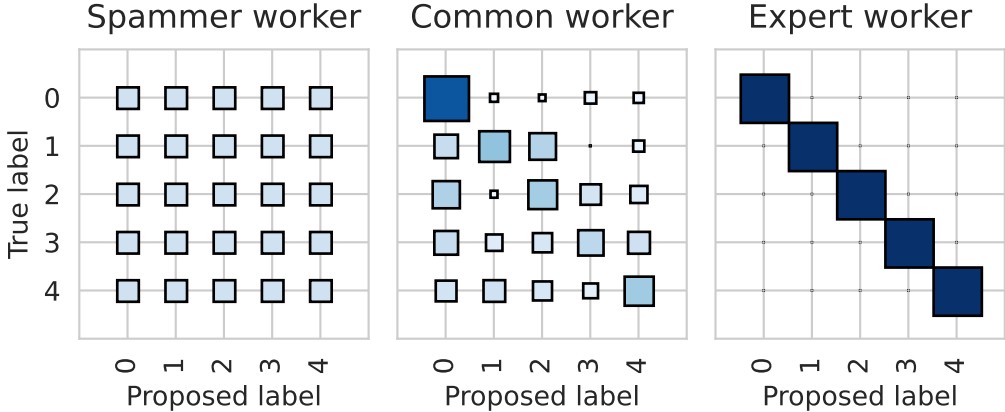

Figure 10: Three types of profiles of worker confusion matrices simulated with `peerannot`. The spammer answers independently of the true label. Expert workers identify classes without mistakes. In practice common workers are good for some classes but might confuse two (or more) labels. All workers are simulated using the `peerannot simulate` command presented in Section 3.2.

In Figure 10, we illustrate multiple workers' profile (as reflected by their confusion matrix) on a simulate scenario where the ground truth is available. For that we generate toy datasets with the `simulate` command from `peerannot`. In particular, we display a type of worker that can hurt

data quality: the spammer. Raykar and Yu (2011) defined a spammer as a worker that answers independently of the true label:

$$\forall k \in [K], \ \mathbb{P}(y_i^{(j)} = k | y_i^\star) = \mathbb{P}(y_i^{(j)} = k) \ . \tag{3}$$

Each row of the confusion matrix represents the label's probability distribution given a true label. Hence, the spammer has a confusion matrix with near-identical rows. Apart from the spammer, common mistakes often involve workers mixing up one or several classes. Expert workers have a confusion matrix close to the identity matrix.

# 4    Learning from crowdsourced tasks

Commonly, tasks are crowdsourced to create a large annotated training set as modern machine learning models require more and more data. The aggregation step then simply becomes the first step in the complete learning pipeline. However, instead of aggregating labels, modern neural networks are directly trained end-to-end from multiple noisy labels.

## 4.1    Popular models

In recent years, directly learning a classifier from noisy labels was introduced. Two of the most used models: CrowdLayer (Rodrigues and Pereira 2018) and CoNAL (Chu, Ma, and Wang 2021), are directly available in `peerannot`. These two learning strategies directly incorporate a DS-inspired noise model in the neural network's architecture.

### 4.1.1    CrowdLayer

CrowdLayer trains a classifier with noisy labels as follows. Let the scores (logits) output by a given classifier neural network $\mathscr{C}$ be $z_i = \mathscr{C}(x_i)$. Then CrowdLayer adds as a last layer $\pi \in \mathbb{R}^{n_{\text{worker}} \times K \times K}$, the tensor of all $\pi^{(j)}$'s such that the crossentropy loss (CE) is adapted to the crowdsourcing setting into $\mathscr{L}_{CE}^{\text{CrowdLayer}}$ and computed as:

$$\mathscr{L}_{CE}^{\text{CrowdLayer}}(x_i) = \sum_{j \in \mathscr{A}(x_i)} \text{CE}\left(\sigma\left(\pi^{(j)}\sigma(z_i)\right), y_i^{(j)}\right) \ ,$$

where the crossentropy loss for two distribution $u, v \in \Delta_K$ is defined as $\text{CE}(u, v) = \sum_{k \in [K]} v_k \log(u_k)$.

Where DS modeled workers as confusion matrices, CrowdLayer adds a layer of $\pi^{(j)}$s into the backbone architecture as a new tensor layer to transform the output probabilities. The backbone classifier predicts a distribution that is then corrupted through the added layer to learn the worker-specific confusion. The weights in the tensor layer of $\pi^{(j)}$s are learned during the optimization procedure.

### 4.1.2    CoNAL

For some datasets, it was noticed that global confusion occurs between the proposed classes. It is the case for example in the `LabelMe` dataset (Rodrigues et al. 2017) where classes overlap. In this case, Chu, Ma, and Wang (2021) proposed to extend the CrowdLayer model by adding global confusion matrix $\pi^g \in \mathbb{R}^{K \times K}$ to the model on top of each worker's confusion.

Given the output $z_i = \mathscr{C}(x_i) \in \mathbb{R}^K$ of a given classifier and task, CoNAL interpolates between the prediction corrected by local confusions $\pi^{(j)}z_i$ and the prediction corrected by a global confusion

$\pi^g z_i$. The loss function is computed as follows:

$$\mathcal{L}_{CE}^{CoNAL}(x_i) = \sum_{j \in \mathscr{A}(x_i)} CE(h_i^{(j)}, y_i^{(j)}) \ ,$$

$$\text{with } h_i^{(j)} = \sigma\left(\left(\omega_i^{(j)}\pi^g + (1-\omega_i^{(j)})\pi^{(j)}\right)z_i\right) \ .$$

The interpolation weight $\omega_i^{(j)}$ is unobservable in practice. So, to compute $h_i^{(j)}$, the weight is obtained through an auxiliary network. This network takes as input the image and worker information and outputs a task-related vector $v_i$ and a worker-related vector $u_j$ of the same dimension. Finally, $w_i^{(j)} = (1 + \exp(-u_j^\top v_i))^{-1}$.

Both CrowdLayer and CoNAL model worker confusions directly in the classifier's weights to learn from the noisy collected labels and are available in `peerannot` as we will see in the following.

## 4.2 Prediction error when learning from crowdsourced tasks

The AccTrain metric presented in Section 3.2 might no longer be of interest when training a classifier. Classical error measurements involve a test dataset to estimate the generalization error. To do so, we present hereafter two error metrics. Assuming we trained our classifier $\mathscr{C}$ on a training set and that there is a test set available with known true labels:

- the test accuracy is computed as $\frac{1}{n_{test}} \sum_{i=1}^{n_{test}} \mathbf{1}_{\{y_i^\star = \hat{y}_i\}}$.
- the expected calibration error (Guo et al. 2017) over $M$ equally spaced bins $I_1, \dots, I_M$ partitioning the interval $[0, 1]$, is computed as:

$$ECE = \sum_{m=1}^{M} \frac{|B_m|}{n_{task}} |acc(B_m) - conf(B_m)| \ ,$$

with $B_m = \{x_i | \mathscr{C}(x_i)_{[1]} \in I_m\}$ the tasks with predicted probability in the $m$-th bin, $acc(B_m)$ the accuracy of the network for the samples in $B_m$ and $conf(B_m)$ the associated empirical confidence. More precisely:

$$acc(B_m) = \frac{1}{|B_m|} \sum_{i \in B_m} \mathbf{1}(\hat{y}_i = y_i^\star) \quad \text{and} \quad conf(B_m) = \frac{1}{|B_m|} \sum_{i \in B_m} \sigma(\mathscr{C}(x_i))_{[1]} \ .$$

The accuracy represents how well the classifier generalizes, and the expected calibration error (ECE) quantifies the deviation between the accuracy and the confidence of the classifier. Modern neural networks are known to often be overconfident in their predictions (Guo et al. 2017). However, it has also been remarked that training on crowdsourced data, depending on the strategy, mitigates this confidence issue. That is why we propose to compare them both in our coming experiments. Note that the ECE error estimator is known to be biased (Gruber and Buettner 2022). Smaller training sets are known to have a higher ECE estimation error. And in the crowdsourcing setting, openly available datasets are often quite small.

## 4.3 Use case with `peerannot` on real datasets

Few real crowdsourcing experiments have been released publicly. Among the available ones, `CIFAR-10H` (Peterson et al. 2019) is one of the largest with 10000 tasks labeled by workers (the testing set of CIFAR-10). The main limitation of `CIFAR-10H` is that there are few disagreements between workers and a simple majority voting already leads to a near-perfect AccTrain error. Hence, comparing the impact of aggregation and end-to-end strategies might not be relevant (Peterson et al.

2019; Aitchison 2021), it is however a good benchmark for task difficulty identification and worker evaluation scoring. Each of these dataset contains a test set, with known ground truth. Thus, we can train a classifier from the crowdsourced data, and compare predictive performance on the test set.

The `LabelMe` dataset was extracted from crowdsourcing segmentation experiments and a subset of $K = 8$ classes was released in Rodrigues et al. (2017).

Let us use `peerannot` to train a VGG-16 with two dense layers on the `LabelMe` dataset. Note that this modification was introduced to reach state-of-the-art performance in (Chu, Ma, and Wang 2021). Other models from the `torchvision` library can be used, such as Resnets, Alexnet *etc.* The `aggregate-deep` command takes as input the path to the data folder, `--output-name/-o` is the name for the output file, `--n-classes/-K` the number of classes, `--strategy/-s` the learning strategy to perform (*e.g.*, CrowdLayer or CoNAL), the backbone classifier in `--model` and then optimization hyperparameters for pytorch described with more details using the `peerannot aggregate-deep --help` command.

Table 4: Generalization performance on LabelMe dataset depending on the learning strategy from the crowdsourced labels. The network used is a VGG-16 with two dense layers for all methods.

| | method | AccTest | ECE |
|---|---|---|---|
| | **Table 4** | | |
| 0 | CoNAL[scale=0] | 81.061 | 0.189 |
| 1 | DS | 85.606 | 0.143 |
| 2 | CrowdLayer | 86.448 | 0.136 |
| 3 | CoNAL[scale=1e-4] | 87.205 | 0.117 |
| 4 | GLAD | 87.542 | 0.124 |
| 5 | NS | 88.468 | 0.115 |
| 6 | MV | 88.889 | 0.112 |

As we can see, CoNAL strategy performs best. In this case, it is expected behavior as CoNAL was created for the `LabelMe` dataset. However, using `peerannot` we can look into **why modeling common confusion returns better results with this dataset**. To do so, we can explore the datasets from two points of view: worker-wise or task-wise in Section 5.

# 5 Identifying tasks difficulty and worker abilities

If a dataset requires crowdsourcing to be labeled, it is because expert knowledge is long and costly to obtain. In the era of big data, where datasets are built using web scraping (or using a platform like Amazon Mechanical Turk), citizen science is popular as it is an easy way to produce many labels.

However, mistakes and confusions happen during these experiments. Sometimes involuntarily (*e.g.,* because the task is too hard or the worker is unable to differentiate between two classes) and sometimes voluntarily (*e.g.,* the worker is a spammer).

Underlying all the learning models and aggregation strategies, the cornerstone of crowdsourcing is evaluating the trust we put in each worker depending on the presented task. And with the gamification of crowdsourcing (Servajean et al. 2016; Tinati et al. 2017), it has become essential to find scoring metrics both for workers and tasks to keep citizens in the loop so to speak. This is the purpose of the identification module in `peerannot`.

Our test cases are both the `CIFAR-10H` dataset and the `LabelMe` dataset to compare the worker and task evaluation depending on the number of votes collected. Indeed, the `LabelMe` dataset has only up to three votes per task whereas `CIFAR-10H` accounts for nearly fifty votes per task.

## 5.1 Exploring tasks' difficulty

To explore the tasks' intrinsic difficulty, we propose to compare three scoring metrics:

- the entropy of the NS distribution: the entropy measures the inherent uncertainty of the distribution to the possible outcomes. It is reliable with a big enough and not adversarial crowd. More formally:

$$\forall i \in [n_{\text{task}}], \text{ Entropy}(\hat{y}_i^{NS}) = - \sum_{k \in [K]} (y_i^{NS})_k \log\left((y_i^{NS})_k\right) \ .$$

- GLAD's scoring: by construction, Whitehill et al. (2009) introduced a scalar coefficient to score the difficulty of a task.
- the Weighted Area Under the Margins (WAUM): introduced by Lefort et al. (2022), this weighted area under the margins indicates how difficult it is for a classifier $\mathscr{C}$ to learn a task's label. This procedure is done with a budget of $T > 0$ epochs. Given the crowdsourced labels and the trust we have in each worker denoted $s^{(j)}(x_i) > 0$, the WAUM of a given task $x_i \in \mathscr{X}$ and a set of crowdsourced labels $\{y_i^{(j)}\}_j \in [K]^{|\mathscr{A}(x_i)|}$ is defined as:

$$\text{WAUM}(x_i) := \frac{1}{|\mathscr{A}(x_i)|} \sum_{j \in \mathscr{A}(x_i)} s^{(j)}(x_i) \left\{ \frac{1}{T} \sum_{t=1}^{T} \sigma(\mathscr{C}(x_i))_{y_i^{(j)}} - \sigma(\mathscr{C}(x_i))_{[2]} \right\} \ ,$$

where we remind that $\mathscr{C}(x_i))_{[2]}$ is the second largest probability output by the classifier $\mathscr{C}$ for the task $x_i$.

The weights $s^{(j)}(x_i)$ are computed à la Servajean et al. (2017):

$$\forall j \in [n_{\text{worker}}], \forall i \in [n_{\text{task}}], \ s^{(j)}(x_i) = \left\langle \sigma(\mathscr{C}(x_i)), \text{diag}(\pi^{(j)}) \right\rangle \ ,$$

where $\hat{\pi}^{(j)}$ is the estimated confusion matrix of worker $w_j$ (by default, the estimation provided by DS).

The WAUM is a generalization of the AUM by Pleiss et al. (2020) to the crowdsourcing setting. A high WAUM indicates a high trust in the task classification by the network given the crowd labels. A low WAUM indicates difficulty for the network to classify the task into the given classes (taking into consideration the trust we have in each worker for the task considered). Where other methods only consider the labels and not directly the tasks, the WAUM directly considers the learning trajectories to identify ambiguous tasks. One pitfall of the WAUM is that it is dependent on the architecture used.

Note that each of these statistics could prove useful in different contexts. The entropy is irrelevant in settings with few labels per task (small $|\mathscr{A}(x_i)|$). For instance, it is uninformative for `LabelMe` dataset. The WAUM can handle any number of labels, but the larger the better. However, as it uses a deep learning classifier, the WAUM needs the tasks $(x_i)_i$ in addition to the proposed labels while the other strategies are feature-blind.

### 5.1.1 CIFAR-1OH dataset

First, let us consider a dataset with a large number of tasks, annotations and workers: the `CIFAR-10H` dataset by Peterson et al. (2019).

```
Unable to display output for mime type(s): text/html
```

Most difficult tasks sorted by class from MV aggregation identified depending on the strategy used (entropy, GLAD or WAUM) using a Resnet34.

```
Unable to display output for mime type(s): text/html
```

The entropy, GLAD's difficulty, and WAUM's difficulty each show different images as exhibited in the interactive Figure. While the entropy and GLAD output similar tasks, in this case, the WAUM often differs. We can also observe an ambiguity induced by the labels in the `truck` category, with the presence of a trailer that is technically a mixup between a `car` and a `truck`.

### 5.1.2  LabelMe dataset

As for the `LabelMe` dataset, one difficulty in evaluating tasks' intrinsic difficulty is that there is a limited amount of votes available per task. Hence, the entropy in the distribution of the votes is no longer a reliable metric, and we need to rely on other models.

Now, let us compare the tasks' difficulty distribution depending on the strategy considered using `peerannot`.

```
Unable to display output for mime type(s): text/html
```

Most difficult tasks sorted by class from MV aggregation identified depending on the strategy used (entropy, GLAD or WAUM) using a VGG-16 with two dense layers.

Note that in this experiment, because the number of labels given per task is in $\{1, 2, 3\}$, the entropy only takes four values. In particular, tasks with only one label all have a null entropy, so not just consensual tasks. The MV is also not suited in this case because of the low number of votes per task.

The underlying difficulty of these tasks mainly comes from the overlap in possible labels. For example, `tallbuildings` are most often found `insidecities`, and so are `streets`. In the `opencountry` we find `forests`, river-coasts and `mountains`.

## 5.2  Identification of worker reliability and task difficulty

From the labels, we can explore different worker evaluation scores. GLAD's strategy estimates a reliability scalar coefficient $\alpha_j$ per worker. With strategies looking to estimate confusion matrices, we investigate two scoring rules for workers:

- The trace of the confusion matrix: the closer to $K$ the better the worker.
- The closeness to spammer metric (Raykar and Yu 2011) (also called spammer score) that is the Frobenius norm between the estimated confusion matrix $\hat{\pi}^{(j)}$ and the closest rank-1 matrix. The further to zero the better the worker. On the contrary, the closer to zero, the more likely it is for the worker to be a spammer. This score separates spammers from common workers and experts (with profiles as in Figure 10).

When the tasks are available, confusion-matrix-based deep learning models can also be used. We thus add to the comparison the trace of the confusion matrices with CrowdLayer and CoNAL on the `LabelMe` datasets. For CoNAL, we only consider the trace of the confusion matrix $\pi^{(j)}$ in the pairwise comparison. Moreover, for CrowdLayer and CoNAL we show in Figure 12 the weights learned without the softmax operation by row to keep the comparison as simple as possible with the actual outputs of the model.

Comparisons in Figure 11 and Figure 12 are plotted pairwise between the evaluated metrics. Each point represents a worker. Each off-diagonal plot shows the joint distribution between the scores of the y-axis row and the x-axis column. They allow us to visualize the relationship between these two variables. The main diagonal represents the (smoothed) marginal distribution of the score of the considered column.

### 5.2.1 CIFAR-10H

The `CIFAR-10H` dataset has few disagreements among workers. However, these strategies disagree on the ranking of good against best workers as they do not measure the same properties.

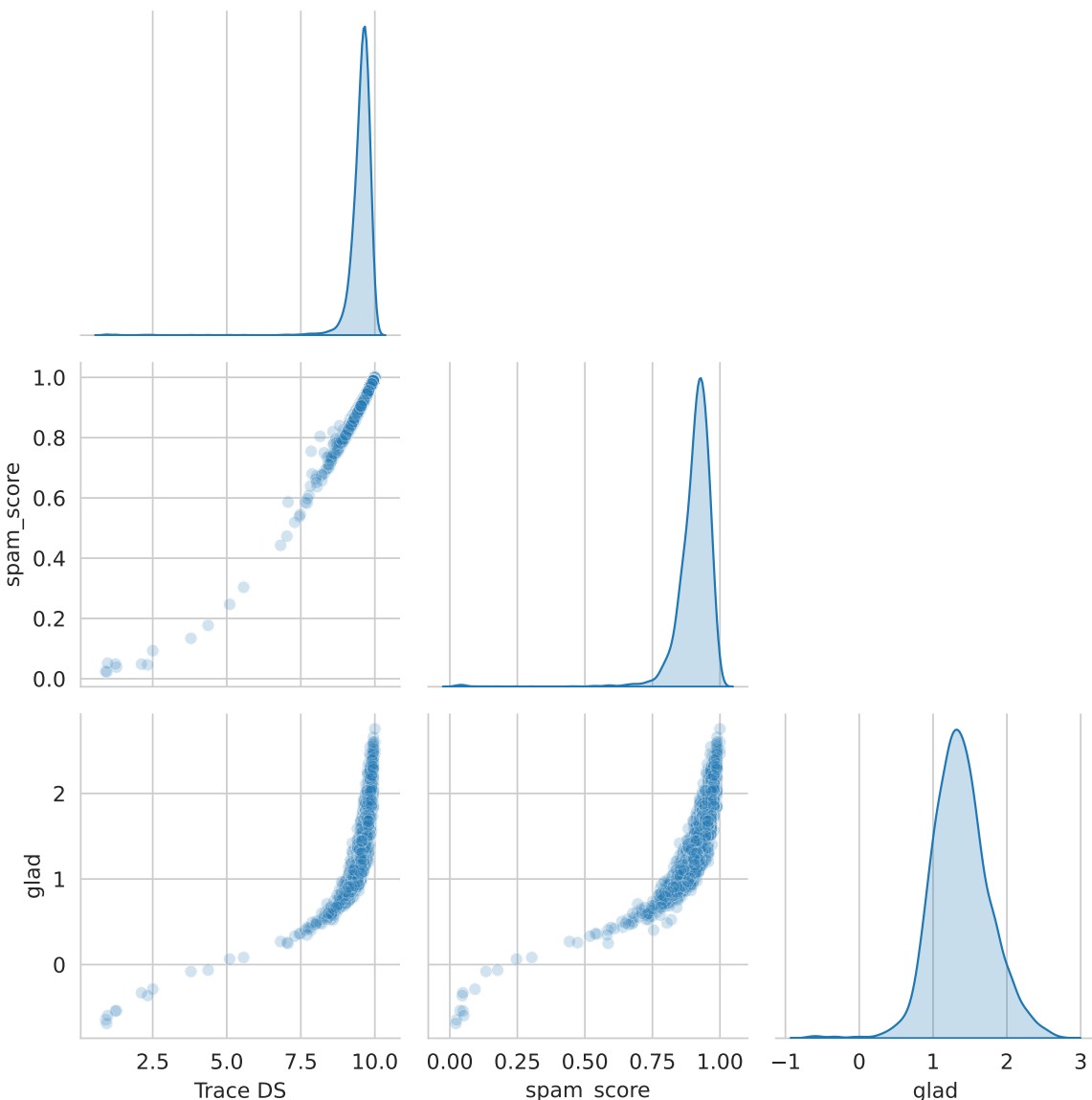

Figure 11: Comparison of ability scores by workers for the CIFAR-10H dataset. All metrics computed identify the same poorly performing workers. A mass of good and expert workers can be seen as the dataset presents few disagreements, thus few data to discriminate expert workers from the otherss.

From Figure 11, we can see that in this dataset, different methods easily separate the worst workers from the rest of the crowd (workers in the left tail of the distribution).

### 5.2.2 LabelMe

Finally, let us evaluate workers for the `LabelMe` dataset. Because of the lack of data (up to 3 labels per task), ranking workers is more difficult than in the `CIFAR-10H` dataset.

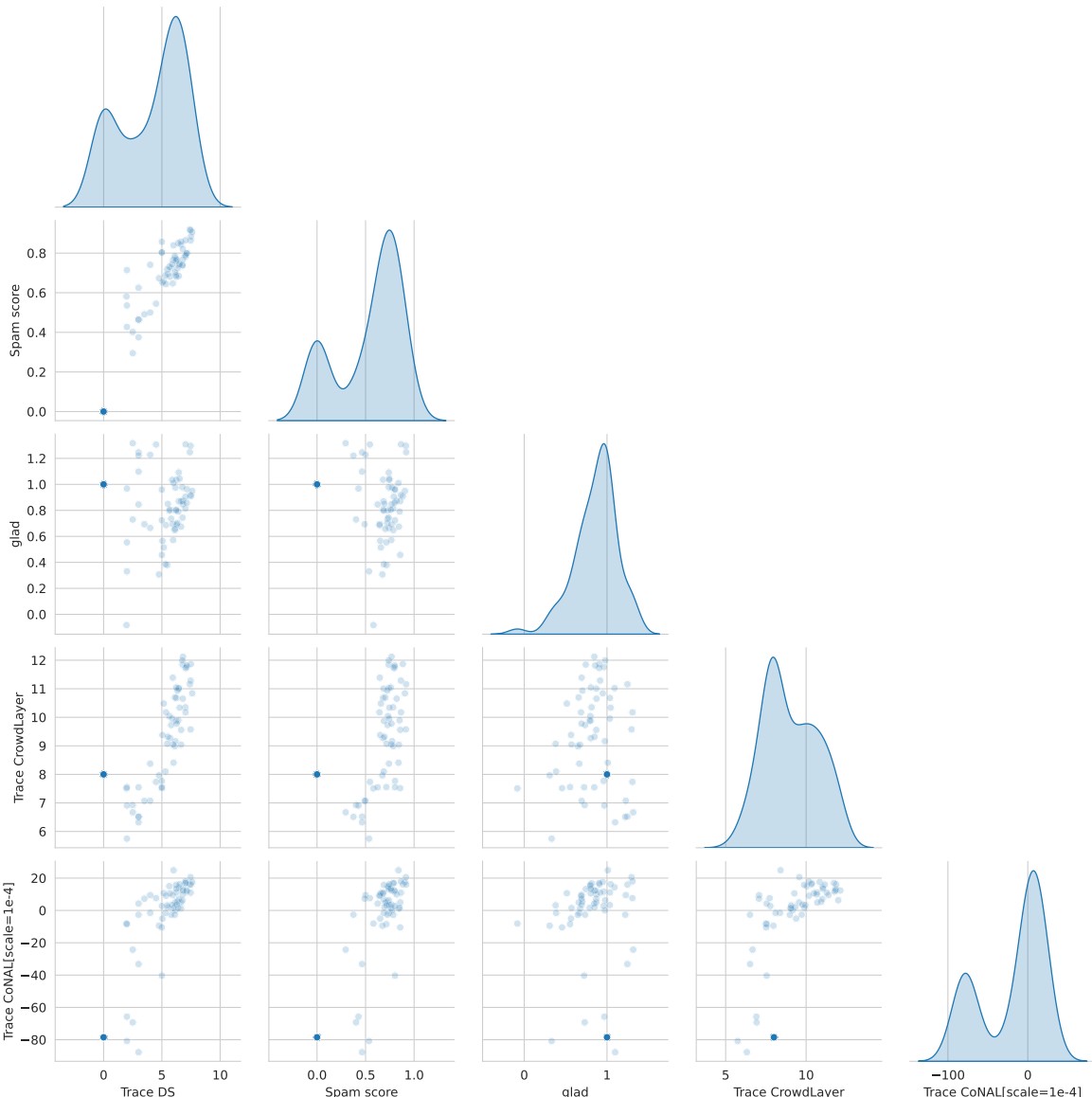

Figure 12: Comparison of ability scores by workers for the LabelMe dataset. With few labels per task, workers are more difficult to rank. It is more difficult to separate workers with their abilities in this crowd. Hence the importance of investigating the generalization performance of the methods presented in the previous section.

We can see in Figure 12 that the number of labels available by task highly impacts the worker evaluation scores. The spam score, DS model and CoNAL all show similar results in the distribution shape (bimodal distribution) whereas GLAD and CrowdLayer are more concentrated. However, this does not account for the ranking of a given worker by the methods considered. The exploration of the dataset lets us look at different scores, but generalization performance presented in Section 4.3 should also be considered in crowdsourcing. This difference in worker evaluation scores indeed further highlights the importance of using multiple test metrics to compare the model's prediction performance in crowdsourcing. Poorly performing workers could be removed from the dataset with

naive strategies like MV or NS. However, some label aggregation strategies like DS or GLAD can sometimes use adversarial votes as information – for example in binary classification, with a worker answering always the opposite label the confusion matrix retrieves the true label. We have seen that the library `peerannot` allows users to explore the datasets, both in terms of tasks and workers, and easily compare predictive performance in this setting.

In practice, the data exploration step can be used to detect possible ambiguities in the dataset's tasks, but also remove answers from spammers to improve the data quality as shown in Figure 1. The easy access to the different strategies allows the user to decide if, for their collected dataset, there is a need for more recent deep-learning-based strategies to improve the results. This is the case for the `LabelMe` dataset. Otherwise, the user can decide that standard aggregation-based crowdsourcing strategies are sufficient and for example, if there are plenty of votes per task like in `CIFAR-10H`, that the entropy of the vote distribution is a criterion that identified enough ambiguous tasks for their case. As often, not a single strategy works best for all datasets, hence the need to perform easy comparisons with `peerannot`.

# 6  Conclusion

We introduced `peerannot`, a library to handle crowdsourced datasets. This library enables both easy label aggregation and direct training strategies with classical state-of-the-art classifiers. The identification module of the library allows exploring the collected data from both the tasks and the workers' point of view for better scorings and data cleaning procedures. Our library also comes with templated datasets to better share crowdsourced datasets. Going beyond templating, it helps the crowdsourcing community to have openly accessible strategies to test, compare and improve to develop common strategies to analyze more and more common crowdsourced datasets.

We hope that this library helps reproducibility in the crowdsourcing community and also standardizes training from crowdsourced datasets. New strategies can easily be incorporated into the open-source code available on GitHub. Finally, as `peerannot` is mostly directed to handle classification datasets, one of our future works would be to consider other `peerannot` modules to handle crowdsourcing for object detection, segmentation and even worker evaluation in other contexts like peer-grading.

# 7  Appendix

## 7.1  Supplementary simulation: Simulated mistakes with discrete difficulty levels on tasks

For an additional simulation setting, we consider the so-called discrete difficulty presented in Whitehill et al. (2009). Contrary to other simulations, we here consider that workers belong to two levels of abilities: `good` or `bad`, and tasks have two levels of difficulty: `easy` or `hard`. The keyword `ratio-diff` indicates the prevalence of each level of difficulty, it is defined as the ratio of `easy` tasks over `hard` tasks:

$$\texttt{ratio-diff} = \frac{\mathbb{P}(\texttt{easy})}{\mathbb{P}(\texttt{hard})} \text{ with } \mathbb{P}(\texttt{easy}) + \mathbb{P}(\texttt{hard}) = 1 \ .$$

Difficulties are then drawn at random. Tasks that are `easy` are answered correctly by every worker. Tasks that are `hard` are answered following the confusion matrix assigned to each worker (as in Section 3.2.1). Each worker then answers independently to the presented tasks.

We simulate $n_{\text{task}} = 500$ tasks and $n_{\text{worker}} = 100$ with 35% of good workers in the crowd and 50% of easy tasks. There are $K = 5$ possible classes. Each task receives $|\mathscr{A}(x_i)| = 10$ labels.

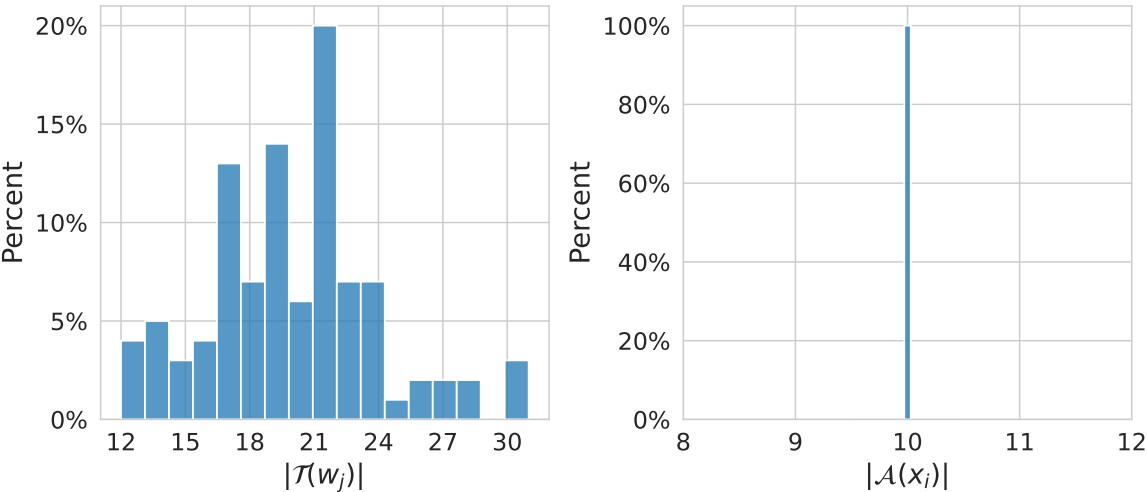

Figure 13: Distribution of the number of tasks given per worker (left) and of the number of labels per task (right) in the setting with simulated discrete difficulty levels.

With the obtained answers, we can look at the aforementioned aggregation strategies performance:

Table 5: AccTrain metric on simulated mistakes made when tasks are associated with a difficulty level considering classical feature-blind label aggregation strategies.

Table 5

|  | MV | GLAD | DS | DSWC[L=2] | DSWC[L=5] | NS |
|---|---|---|---|---|---|---|
| AccTrain | 0.780 | 0.845 | 0.810 | 0.600 | 0.660 | 0.790 |

Finally, in this setting involving task difficulty coefficients, the only strategy that involves a latent variable for the task difficulty, knowing GLAD, outperforms the other strategies (see Table 5). Note that in this case, creating clusters of answers leads to worse decisions than an MV aggregation.

## 7.2 Comparison with other libraries

In this section, we provide several comparisons with the Ustalov, Pavlichenko, and Tseitlin (2023) library.

- Framework: `peerannot` focuses on image classification problems with categorical answers. `crowd-kit` also considers textual responses and image segmentation with three aggregation strategies for each field.
- Data storage: `peerannot` introduces this `.json` storage that can handle large datasets. `crowd-kit` stores the collected data in a `.csv` file with columns `task`, `worker`, `label`.
- Identification module: one of the major differences between the two libraries resides in the `identification` module of `peerannot`. This module allows us to explore the dataset and detect poorly performing workers / difficult tasks easily. `crowd-kit` only allows us to explore workers with the `accuracy_on_aggregation` metric that computes the accuracy of a worker given aggregated hard labels. `peerannot`, as demonstrated in Section 5, proposes several metrics such as the spam score, GLAD's worker ability coefficient and the trace of the

confusion matrices. As for the task side, `peerannot` proposes the different popular metrics in `crowd-kit` accompanied with the WAUM (and also the AUMC) metrics from Lefort et al. (2022) and GLAD's difficulty coefficients.

- Training: `peerannot` lets users directly `train` a neural network architecture from the aggregated labels. This feature is not proposed by `crowd-kit`.
- Simulation: `peerannot` created a `simulate` module to check strategies on. This feature is also not in the `crowd-kit` library.

Finally, to compare different strategies across libraries, we implemented a crowdsourcing benchmark in the Benchopt (Moreau et al. (2022)) library. The Benchopt library allows users to easily compare and reproduce optimization problem benchmarks between multiple frameworks. After running each strategy, we measure the cumulated time taken to reach the optimum during the optimization steps. The metric measured on the y-axis is the AccTrain. Each strategy is run 5 times until convergence. The differences in results across iterations for the MV strategy come from the randomness in the choice in case of equalities. We provide a clone of the crowdsourcing benchmark and the results are obtained by running the following command:

```
benchopt run ./benchmark_crowdsourcing
```

First, let us see the performance on the Bluebirds dataset, a small dataset with 39 workers, 108 tasks and $K = 2$ classes.

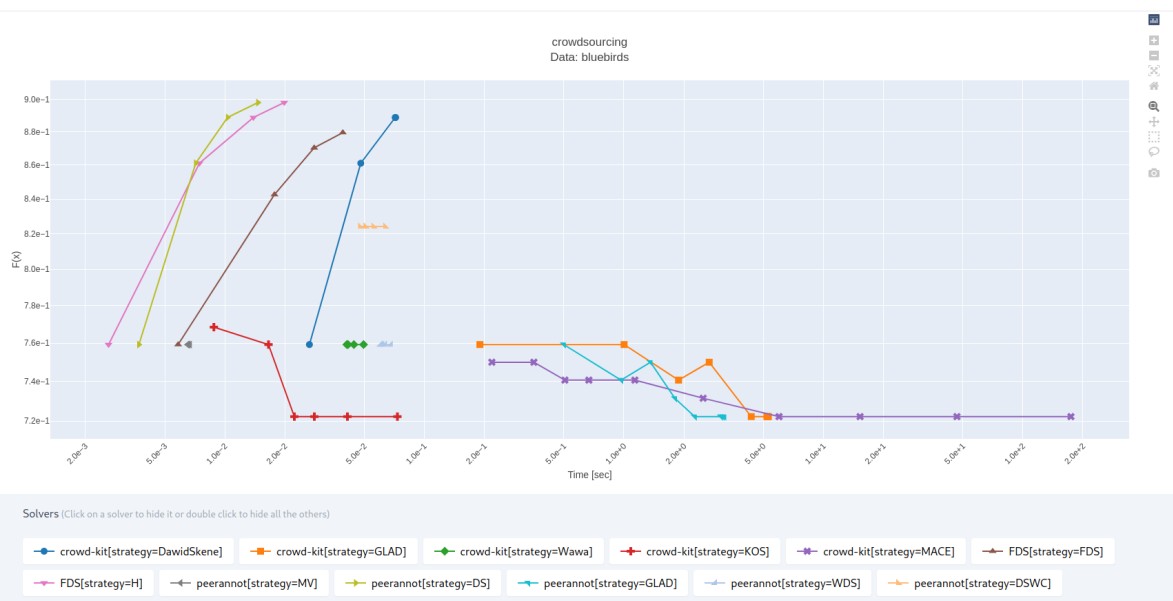

Figure 14: Aggregation strategies computational time during optimization procedure for the BlueBirds dataset with K=2.

We see in Figure 14 that the DS strategy from `peerannot` is the first to reach the optimum, followed by the Fast-DS strategy and then `crowd-kit` DS. Other strategies do not lead to better accuracy on this dataset and DS seems to be the best fitting strategy.

For the LabelMe dataset, DS strategy is also the best aggregation strategy, faster for `crowd-kit`. The sensitivity of GLAD's method to the priors on $\alpha$ and $\beta$ parameters can lead to large performance differences for real datasets as we see in Figure 15. Note that `crowd-kit`'s KOS strategy is not available for this dataset as it is only made for binary classification datasets.

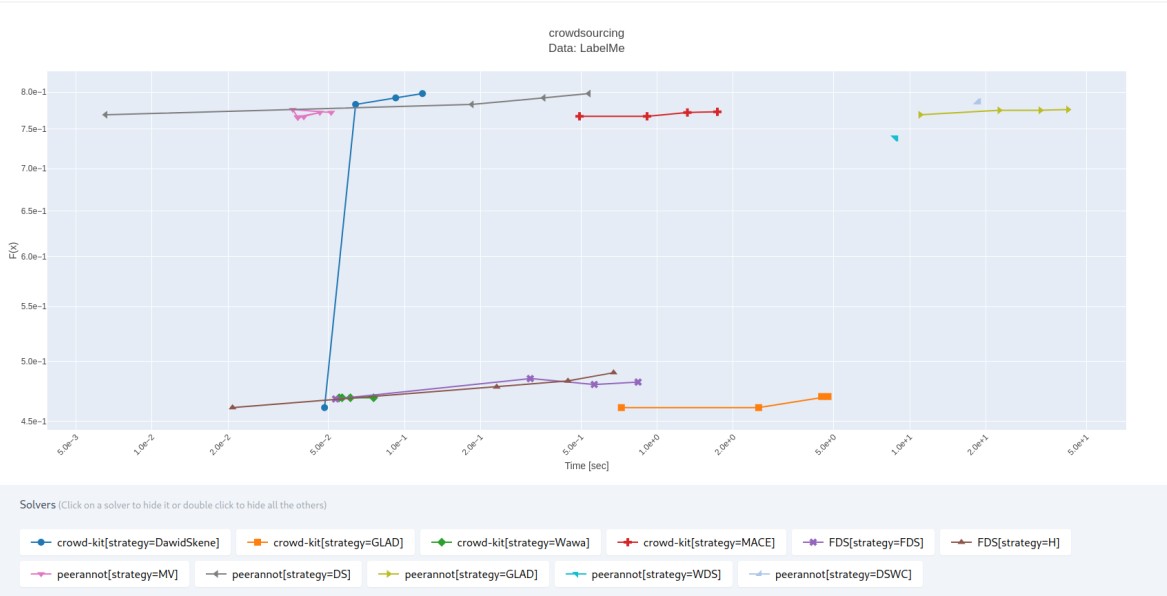

Figure 15: Aggregation strategies computational time during optimization procedure for the LabelMe dataset with K=8

## 7.3 Examples of images in CIFAR-10H and Labelme

In this section, we provide examples of images from the `CIFAR-10H` and `LabelMe` datasets. Both of these datasets came with known true labels. For `CIFAR-10H`, the true labels were from the original `CIFAR-10` dataset. For `LabelMe`, the true labels were determined by the authors at release.

## 7.4 Case study with bird sound classification

We shared our results on the classical CIFAR-10H and LabelMe datasets. More recently, Lehikoinen et al. (2023) developed a platform for bird sound classification. They released the data for the following crowdsourcing experiment. Given the sample of the audio of a species (denoted as a letter on their web portal), users were presented with a new audio sample (the candidate). The question is as follows: *Is the species vocalizing in the candidate the same as the species in the letter¿* 'The answer is a binary yes or no. In total, $n_{\mathrm{worker}} = 205$ workers labeled $n_{\mathrm{task}} = 79\,592$ candidates. Each task received between 1 and 77 annotations. Workers answered between 1 and 30 759 tasks (only one worker achieved that record, and 23% of the workers answered 100 tasks). There is no test set available as is in the original dataset. However, to have an idea of the level of performance of the label aggregation strategies, we use the fact that workers reported their level of expertise between 1 and 4. The latter corresponds to "I am bird researcher or professional birdwatcher". This generates a test set of 13 041 tasks where the expert label is used as the current truth. This test set is only used to compute the AccTrain metric. Note that we do not perform deep-learning methods as the tasks of comparing the birds from two audio files and designing specific architectures to match this framework is out of the scope of this paper.

We then can run our aggregation strategies, and from Table 6 we see that strategies reach the same levels of label recovery, however naive they are. Indeed, most tasks have very few disagreements. Note that NS and MV performance difference comes from the random tie-breakers in case of equalities.

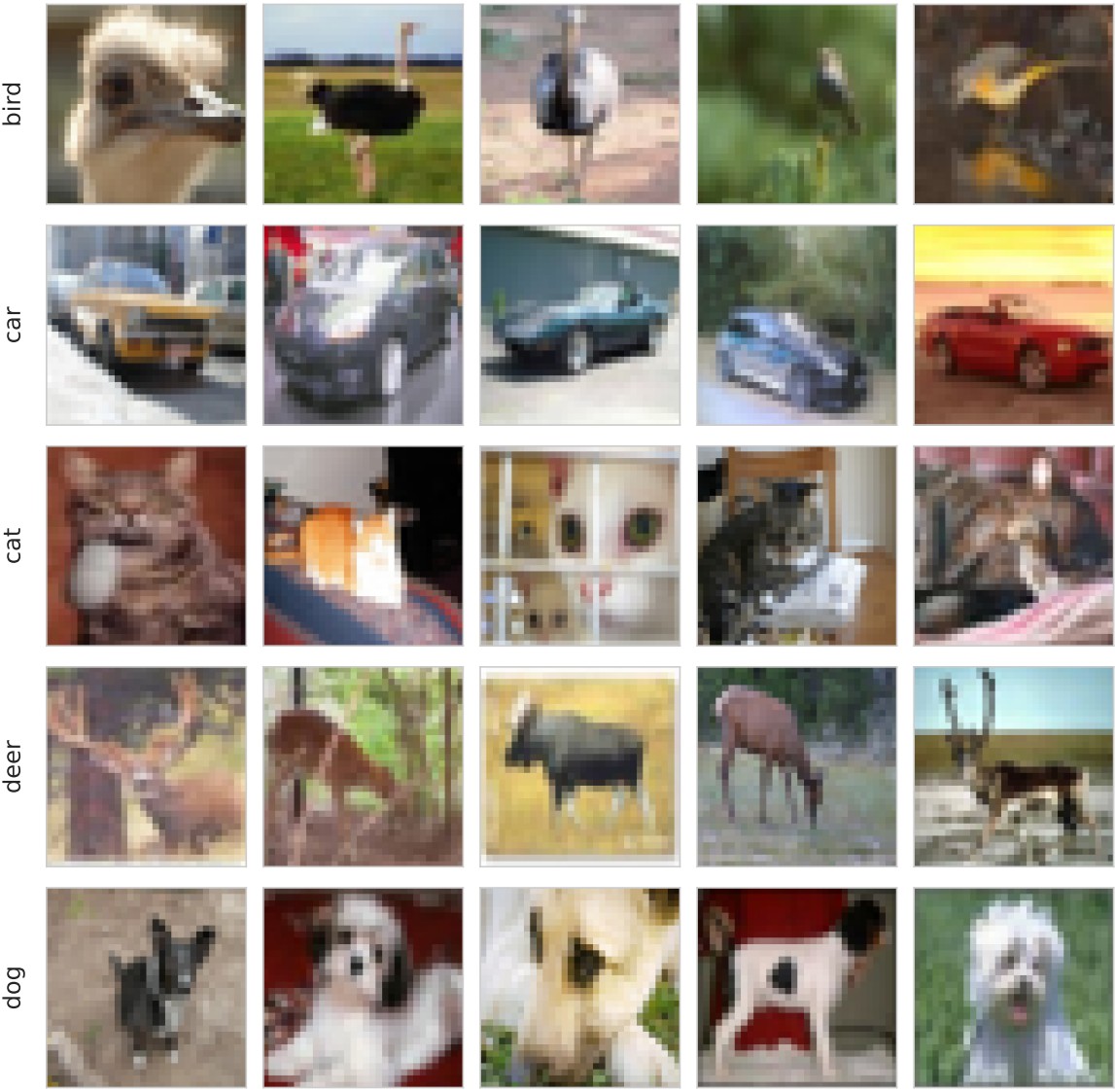

Figure 16: Example of images from CIFAR-10H. We display images row-wise according to the true label given initially in CIFAR-10.

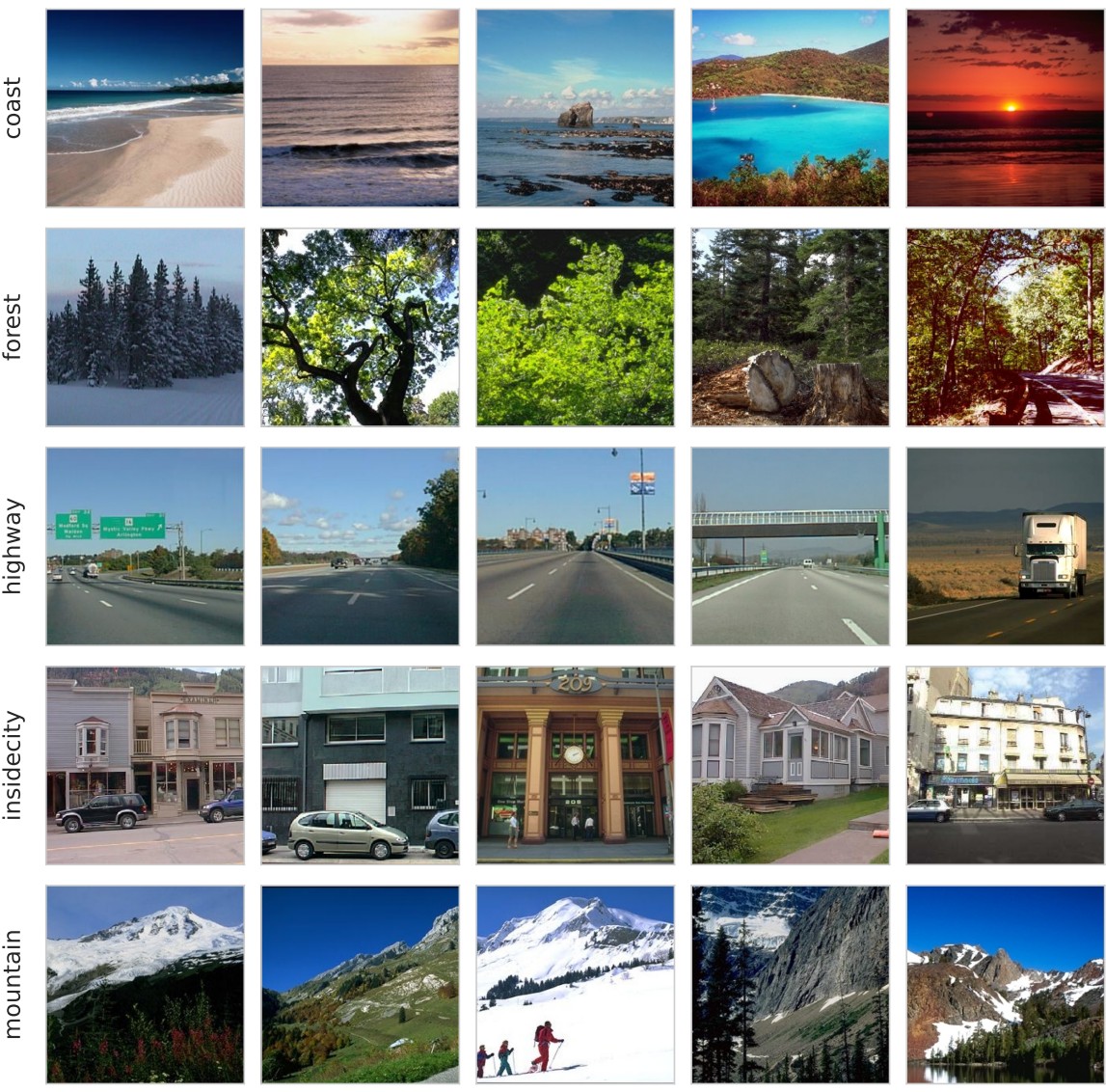

Figure 17: Example of images from LabelMe. We display images row-wise according to the true label given with the crowdsourced data.

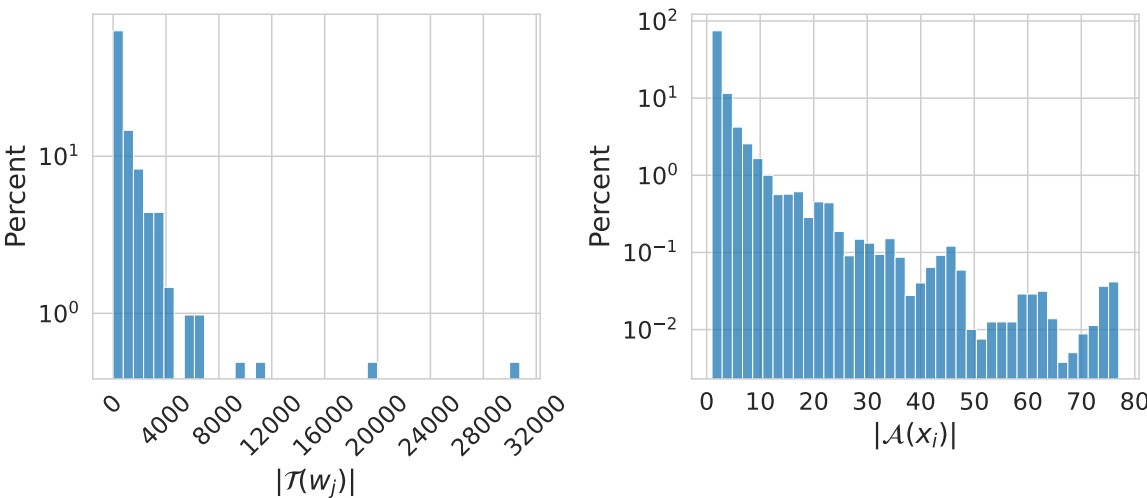

Figure 18: Distribution of the number of tasks given per worker (left) and of the number of labels per task (right) in the Audio Birds letters dataset.

Table 6: AccTrain metric on birds audio dataset considering classical feature-blind label aggregation strategies.



Table 6

|          | MV    | DS    | GLAD  | NS    |
|----------|-------|-------|-------|-------|
| AccTrain | 0.954 | 0.946 | 0.950 | 0.960 |



We can explore what tasks lead to the most disagreements depending on the entropy criterion or GLAD's difficulty-estimated latent variable.

Using the entropy criterion, the most difficult tasks (highest entropy) and GLAD's difficulty, we recover the index of the most ambiguous tasks.

```
Highest entropy tasks index: [28272, 25827, 40989, 2771, 55559]
Highest GLAD difficulty index: [2347, 435, 8710, 8992, 51700]
```

- Entropy: we obtain the candidate `MRG18_20180514_000000_203.mp3` that was to be compared with the letter `HLO15_20180515_021439_31.mp3` (one worker agrees and another disagrees):

./datasets/birds_audio/bird_sound_training_data/audio_files/MRG18_20180514_000000_203.mp3

./datasets/birds_audio/bird_sound_training_data/audio_files/HLO15_20180515_021439_31.mp3

And the candidate `MRG24_20180512_000000_437.mp3` that was to be compared with the letter `HLO12_20180511_150153_42.mp3` (one worker agrees and another disagrees):

./datasets/birds_audio/bird_sound_training_data/audio_files/MRG24_20180512_000000_437.mp3

./datasets/birds_audio/bird_sound_training_data/audio_files/HLO12_20180511_150153_42.mp3

- GLAD: we obtain the candidate `HLO04_20180511_034424_15.mp3` that was to be compared with the letter `MRG11_20180519_000000_506.mp3` (53 votes, 29 aggreeing and 24 disagreeing):

./datasets/birds_audio/bird_sound_training_data/audio_files/HLO04_20180511_034424_15.mp3

And the candidate `MRG27_20180512_000000_597.mp3` that was to be compared with the letter `HLO01_20180601_080126_30.mp3` (43 votes, 23 aggreeing and 20 disagreeing):

In this dataset, a single task with two different votes has the highest entropy. GLAD's coefficient lets us explore tasks with multiple votes where workers were split.

We can also explore the dataset from a worker's point of view and visualize workers' performance and how many are identified as poorly performing. This gives us an idea of the level of noise in the answers.

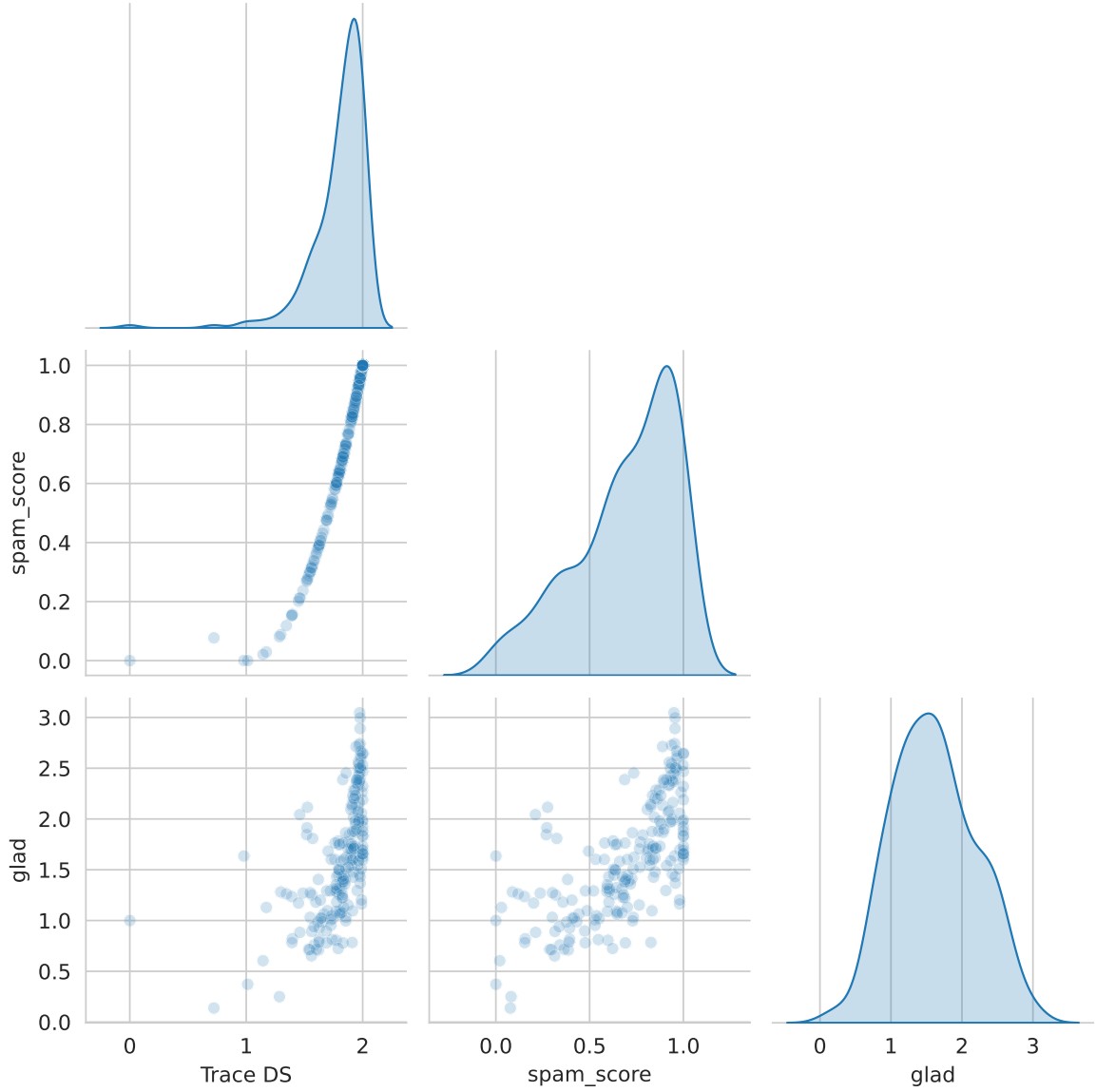

Figure 19: Comparison of ability scores by workers for the birds audio dataset. Most workers do seem to perform similarly, with very little noise voluntarily induced.

From Figure 19, we notice that very few workers are identified as spammers and that different

worker identification strategies seem to perform similarly. Here we show the worse workers' indices depending on each strategy.

```
Worse workers using GLAD [94, 80, 109, 35, 45]
Worse workers using DS trace [69, 94, 172, 109, 35]
Worse workers using Spam Score [69, 109, 172, 35, 130]
```

One of the closing statements of Lehikoinen et al. (2023) is "we learned lessons for how to better implement similar citizen science projects in the future". On one hand, identifying the most ambiguous tasks can help by saving only these tasks to the most expert workers and acquiring better data. On the other hand, combining the task difficulty with the worker ability performance metrics could help to create personal feeds of tasks to label and generate more worker participation. Finally, the label aggregation step can lead to training classifiers with better labels. We hope that allowing easy access thanks to the peerannot library to each of those steps can indeed help to better implement citizen science projects and use the collected data.

Aitchison, L. 2021. "A Statistical Theory of Cold Posteriors in Deep Neural Networks." In *ICLR*.

Cao, P, Y Xu, Y Kong, and Y Wang. 2019. "Max-MIG: An Information Theoretic Approach for Joint Learning from Crowds." In *ICLR*.

Chagneux, M, S LeCorff, P Gloaguen, C Ollion, O Lepâtre, and A Bruge. 2023. "Macrolitter Video Counting on Riverbanks Using State Space Models and Moving Cameras." *Computo*, February. https://computo.sfds.asso.fr/published-202301-chagneux-macrolitter.

Chu, Z, J Ma, and H Wang. 2021. "Learning from Crowds by Modeling Common Confusions." In *AAAI*, 5832–40.

Dawid, AP, and AM Skene. 1979. "Maximum Likelihood Estimation of Observer Error-Rates Using the EM Algorithm." *J. R. Stat. Soc. Ser. C. Appl. Stat.* 28 (1): 20–28.

Deng, J., W. Dong, R. Socher, L.-J. Li, K. Li, and L. Fei-Fei. 2009. "ImageNet: A Large-Scale Hierarchical Image Database." In *CVPR*.

Gao, G, and D Zhou. 2013. "Minimax Optimal Convergence Rates for Estimating Ground Truth from Crowdsourced Labels." *arXiv Preprint arXiv:1310.5764*.

Garcin, C., A. Joly, P. Bonnet, A. Affouard, J.-C. Lombardo, M. Chouet, M. Servajean, T. Lorieul, and J. Salmon. 2021. "Pl@ntNet-300K: A Plant Image Dataset with High Label Ambiguity and a Long-Tailed Distribution." In *Proceedings of the Neural Information Processing Systems Track on Datasets and Benchmarks*.

Gruber, S G, and F Buettner. 2022. "Better Uncertainty Calibration via Proper Scores for Classification and Beyond." In *Advances in Neural Information Processing Systems*.

Guo, C, G Pleiss, Y Sun, and KQ Weinberger. 2017. "On Calibration of Modern Neural Networks." In *ICML*, 1321.

Imamura, H, I Sato, and M Sugiyama. 2018. "Analysis of Minimax Error Rate for Crowdsourcing and Its Application to Worker Clustering Model." In *ICML*, 2147–56.

James, GM. 1998. "Majority Vote Classifiers: Theory and Applications." PhD thesis, Stanford University.

Kasmi, G, Y-M Saint-Drenan, D Trebosc, R Jolivet, J Leloux, B Sarr, and L Dubus. 2023. "A Crowdsourced Dataset of Aerial Images with Annotated Solar Photovoltaic Arrays and Installation Metadata." *Scientific Data* 10 (1): 59.

Khattak, FK. 2017. "Toward a Robust and Universal Crowd Labeling Framework." PhD thesis, Columbia University.

Krizhevsky, A, and G Hinton. 2009. "Learning Multiple Layers of Features from Tiny Images." University of Toronto.

Lefort, T, B Charlier, A Joly, and J Salmon. 2022. "Identify Ambiguous Tasks Combining Crowdsourced Labels by Weighting Areas Under the Margin." *arXiv Preprint arXiv:2209.15380*.

Lehikoinen, P., M. Rannisto, U. Camargo, A. Aintila, P. Lauha, E. Piirainen, P. Somervuo, and O. Ovaskainen. 2023. "A Successful Crowdsourcing Approach for Bird Sound Classification." *Citizen Science: Theory and Practice* 8 (1): 16. https://doi.org/10.5334/cstp.556.

Lin, Tsung-Yi, Michael Maire, Serge J. Belongie, Lubomir D. Bourdev, Ross B. Girshick, James Hays, Pietro Perona, Deva Ramanan, Piotr Dollá r, and C. Lawrence Zitnick. 2014. "Microsoft COCO: Common Objects in Context." *CoRR* abs/1405.0312. http://arxiv.org/abs/1405.0312.

Marcel, S, and Y Rodriguez. 2010. "Torchvision the Machine-Vision Package of Torch." In *Proceedings of the 18th ACM International Conference on Multimedia*, 1485–88. MM '10. New York, NY, USA: Association for Computing Machinery.

Moreau, Thomas, Mathurin Massias, Alexandre Gramfort, Pierre Ablin, Pierre-Antoine Bannier, Benjamin Charlier, Mathieu Dagréou, et al. 2022. "Benchopt: Reproducible, Efficient and Collaborative Optimization Benchmarks." In *NeurIPS*. https://arxiv.org/abs/2206.13424.

Park, Seo Yeon, and Cornelia Caragea. 2022. "On the Calibration of Pre-Trained Language Models Using Mixup Guided by Area Under the Margin and Saliency." In *ACML*, 5364–74.

Passonneau, R J., and B Carpenter. 2014. "The Benefits of a Model of Annotation." *Transactions of the Association for Computational Linguistics* 2: 311–26.

Paszke, A, S Gross, F Massa, A Lerer, J Bradbury, G Chanan, T Killeen, et al. 2019. "PyTorch: An Imperative Style, High-Performance Deep Learning Library." In *NeurIPS*, 8024–35.

Peterson, J C., R M. Battleday, T L. Griffiths, and O Russakovsky. 2019. "Human Uncertainty Makes Classification More Robust." In *ICCV*, 9617–26.

Pleiss, G, T Zhang, E R Elenberg, and K Q Weinberger. 2020. "Identifying Mislabeled Data Using the Area Under the Margin Ranking." In *NeurIPS*.

Raykar, V C, and S Yu. 2011. "Ranking Annotators for Crowdsourced Labeling Tasks." In *NeurIPS*, 1809–17.

Rodrigues, F, M Lourenco, B Ribeiro, and F C Pereira. 2017. "Learning Supervised Topic Models for Classification and Regression from Crowds." *IEEE Transactions on Pattern Analysis and Machine Intelligence* 39 (12): 2409–22.

Rodrigues, F, and F Pereira. 2018. "Deep Learning from Crowds." In *AAAI*. Vol. 32.

Rodrigues, F, F Pereira, and B Ribeiro. 2014. "Gaussian Process Classification and Active Learning with Multiple Annotators." In *ICML*, 433–41. PMLR.

Servajean, M, A Joly, D Shasha, J Champ, and E Pacitti. 2016. "ThePlantGame: Actively Training Human Annotators for Domain-Specific Crowdsourcing." In *Proceedings of the 24th ACM International Conference on Multimedia*, 720–21. MM '16. New York, NY, USA: Association for Computing Machinery.

———. 2017. "Crowdsourcing Thousands of Specialized Labels: A Bayesian Active Training Approach." *IEEE Transactions on Multimedia* 19 (6): 1376–91.

Sinha, V B, S Rao, and V N Balasubramanian. 2018. "Fast Dawid-Skene: A Fast Vote Aggregation Scheme for Sentiment Classification." *arXiv Preprint arXiv:1803.02781*.

Tinati, R, M Luczak-Roesch, E Simperl, and W Hall. 2017. "An Investigation of Player Motivations in Eyewire, a Gamified Citizen Science Project." *Computers in Human Behavior* 73: 527–40.

Ustalov, Dmitry, Nikita Pavlichenko, and Boris Tseitlin. 2023. "Learning from Crowds with Crowd-Kit." arXiv. https://arxiv.org/abs/2109.08584.

Whitehill, J, T Wu, J Bergsma, J Movellan, and P Ruvolo. 2009. "Whose Vote Should Count More: Optimal Integration of Labels from Labelers of Unknown Expertise." In *NeurIPS*. Vol. 22.

Yasmin, R, M Hassan, J T Grassel, H Bhogaraju, A R Escobedo, and O Fuentes. 2022. "Improving Crowdsourcing-Based Image Classification Through Expanded Input Elicitation and Machine Learning." *Frontiers in Artificial Intelligence* 5: 848056.

Zhang, H, M Cissé, Y N. Dauphin, and D Lopez-Paz. 2018. "Mixup: Beyond Empirical Risk Minimization." In *ICLR*.

