# OpenReview forum: "Peerannot: classification for crowdsourced image datasets with Python"
_Computo — Accepted by Computo_

### Review · Reviewer_JvrT · 2023-10-24

**Summary Of Contributions:**

This very clear, well-constructed and documented article details methods for pre-processing crowd-sourced annotation for training classification models.

The authors publish an easy-to-use open source library implementing the algorithms described. They include a few datasets with examples.

They also propose a convention for structuring model training datasets, thus improving the compatibility of tools with different models.

This work is in the spirit of open research. By publishing a practical library and seeking to homogenize practices, the authors are helping to facilitate the work of other players in the community. It's a useful piece of work, and a worthy modern approach.

**Changes And Questions:**

All remarks below are required.
The first point could be addressed by arguing the need to provide the graphics code in the article flow.

## Python Code not very useful in the article

In general, I find most python code in the paper not bringing much value : It does not demonstrate usage of API of the library, but rather some standard matplotlib code to produce the graphs. I feel that the graphs speak from themselves and the readers don't really need to know the code that produced it at first. For the few Python code using internal function of the library, you may put this in some supplementary materials, or as sample code in the library and refer to it in the paper.

I feel it may ease the readability of the paper.

This remark does not apply to CLI /  Bash commands, which I find useful.

## Code formatting and long lines

Several long lines are cut in the PDF output, both for Python and CLI commands.
Examples :
- **2.2 - Page 6** : `"plane", 8 ... `
- **2.2 - Page 7 - end** : `"opencountry", ...`
- **3. - Page 11 - end**:  `--feedback=...`
- ... please double-check for other code (CLI & Python)

## Missing graphs

- **5.1.1 & 5.1.2 - Page 32** : The following text appears 3 times instead of a proper graph :
 `Unable to display output for mime type(s): text/html`

## Misc remarks
- **Figure 6 - Page 13** : Please label the vertical axis "ground truth", or similar
- **Figure 13** : Please explain more in detail what a pair plot is / how to read it. The top-left plot seem to lack a vertical axis / labeling
- **6 Conclusion - Page 37**  : The conclusion is attached to the bibliography, without proper space or title to separate them.

**Comments On Reproducibility:**

The library is well packaged and works perfectly out of the box.
The CLI commands are very clear and online documentation (`--help`) is very good.

Here are some minor suggestions :

* **Enhance the visibility of the documentation**: Googling **peerannot** returns the personal page of the first author and the [**pip** page](https://pypi.org/project/peerannot/). The latter lacks a link to the homepage (either Github project or github.io page) and extended description.

* **Enforce python version compatibility** : I was able to run the library using **python3.10**, but was not sure which version is supported. You may enforce the min/max versions of python in the documentation and [setup config](https://packaging.python.org/en/latest/guides/distributing-packages-using-setuptools/#python-requires)

* **Cifar**: It was not clear in the documentation (but more clear in the paper) that the **cifar10H** folder was provided in the folder **datasets** and required to clone the repository. I first understood I had to download & install it myself locally from official Cifar website. Could you make it more obvious in the documentation ?

* **job_xx.py** : The folder `datasets` contains several **job_x.py** that seem to be private files, not used, referring to private absolute home folder the author.

* **Heavy dependencies** : The dependencies are quite heavy, especially *vision. It may be useful to make torch* dependencies optional, for those only waiting to use `aggregate` &  `identify` only and having already their own training pipeline. Don't bother to do it if the code is too much dependent on this though.

**Reproducibility:**

Yes

**Strengths And Weaknesses:**

The library is particularly well-developed and packaged. Documentation is clear (both online and in the commands themselves). It works perfectly.

The paper is also very clear and well constructed. It does, however, suffer from a few formatting problems and some bloat in the part including code.

---

> ### Author Response · Authors · 2023-11-23
>
> First, we would like the reviewer for the valuable feedback. Below, we address most points of concern.
>
> - **Python code and formatting**: we now moved most of the code for graphics generation in a `utils.py` to help readability. However, and this goes along with the code formatting and the missing paragraphs remarks, to our understanding, the computo final paper is the html webpage and not the pdf document. In the webpage, we hide the non-library related code and display the code related to the API/CLI. The missing Figures written "Unable to display" are actually interactive Figures using Plotly, and those can only be generated and manipulated using the deployed generated webpage currently available at `https://tanglef.github.io/computo_2023.html` per Computo template recommendation.
>
> - **Figure 5**: The vertical axis is now annotated as the true label
> - **Figure 9**: the seaborn pairplot shows the crossed distribution of considered computed metrics. So in the case of CIFAR-10H dataset for workers identification, a point is a worker and on the x-axis we read the score for a metric and on the y-axis another one. The diagonal displays the distribution of the metric of the considered column. So in Fig.11, the top-left graph represents the smoothed distribution of the (matrix) trace of the DS strategy confusion matrices per worker. There is no top-left label as it is not a crossed distribution. We added more explainations about the pairplots in the paper to ease readers.
> - **Conclusion and bibliography**: this is an issue of the pdf paper template from computo, the html webpage doesn't have this problem.
>
> About minor suggestions:
> - **Google**: the pypi page has been updated with longer description, and a meta tag was added and registered in the google console, time is needed for indexation though.
> - **Python version**: we added a badge and tests on Ubuntu for Python 3.8,9 and 10, a min version is added is the `setup.cfg` file
> - The shell files to run experiments have been removed, thank you for noticing them
> - **Dependencies**: Peerannot has 4 main modules: `aggregate`, `aggregate-deep`, `identify` and `train`. Only the `aggregate` one is independent of deep-learning. `Identify` module has the AUM/WAUM metrics that need a neural network / loading the images, the `train` and `aggregate-deep` modules of course use pytorch and torchvision. We highly encourage users to think about the whole training pipeline with their data, especially the `identify` step, and not only the aggregation step. Hence our choice not to make a different (sub-)library for the aggregations and hence they should share the same dependencies.
>
> If there are any other concerns, please let us know and we will do our best to respond to them.

---

> ### Author Response · Authors · 2023-11-23
> **webpage unavailable**
>
> Thanks for your answer.
> I am still unable to access the page :
> https://tanglef.github.io/computo_2023.html

---

> ### Author Response · Authors · 2023-11-23
>
> Please excuse us, there is no .html at the end of the link: https://tanglef.github.io/computo_2023/

---

> ### Comment · Reviewer_JvrT · 2023-12-06
> **Approval**
>
> Thank you for your replies to my comments.
>
> They address all my points.
>
> I have no further comments to make, and I approve publication of the article as it stands.

---

### Review · Reviewer_ZPNA · 2023-11-17

**Summary Of Contributions:**

This article depicts a new python package released by the authors to analyze crowdsourced annotations in the context of (image) classification.
The article gives an overview of existing methods to handle such data, especially the way of aggregating the classifications given by the different workers (persons that annotates).

**Broader Impact Concerns:**

I have no concern about any ethical problem. However, as said above, I am convinced that the authors could find a more relevant use case to show the potential interests of the crowdsourced data problematic.

**Changes And Questions:**

## Major remarks

- In section 2.1, the authors well depict the required dataset format for the package, and especially the `.json` file has depicted in Figure 2. They say that it is more memory friendly than a full matrix where -1 would code missing values. However, I was wondering if the package propose a function to encode a `.json` file in the right format from such a matrix (which is probably a current way of doing such annotations in this field)?
- In section 3.1.3 the authors write the likelihood in the context of known labels, whereas they wrote before that the *with crowdsourced data, the ground truth of a task [\dots] is unknown*, thus I think the authors should focus on the likelihood in this case (case 2 of the original paper of Dawid and Skene's). From a didactic point of view, writing this likelihood and then explaining that strategies like EM algorithm are relevant for optimization (which is not necessary whan the ground truth is known) in this case seems to be in a better order for me.
- On section 3.2, the authors linger for a long time over simulation under different scenarios to (from what I understand) i) depicts their simulation function; ii) discuss in which scenario the different aggregation strategy should be appropriate. I understand that the authors show this feature of the package, but in my opinion, it would be more interesting to work with athe real data sets studied by the authors (as they have a ground truth), then to show the accuracy table on this data set, and to discuss, depending on the best aggregation strategy, what it tells us about the annotation scheme (it highlights that their might be cluster of worker, for instance...). I found the exercise of finding that the best model is the one from which simulations were made a bit long here.
- Section 4.1.1. The quantities $\pi^{(j)}$ are the confusion matrices of each worker, but again, this imposes that the ground truth is known, isn't it?
Again, section 4.2 poses on the case where ground truth is known. I find this a bit confusing as a reader, maybe the authors should clearly reorganize the paper, having  a section that clearly covers the case of crowdsourcing with available ground truth and a one without available ground truth. As most of the sections rely at some points on confusion matrix, I suggest that the authors do not introduce the paper by emphasizing the absence of ground truth on crowdsourced data sets.
- Section 4.1.2. The global confusion matrix should impose to aggregate the workers prediction, how should it be done (relatively to previous section)?
- Header of section 5 is a bit redundant with the introduction.
- End of 5.2.1, is there any recommandation on what to do with the "bad" workers? Is there any alternative to simple discard?
- On the example codes shown by the authors (that all ran on my laptop without problem), such as the ones for simulations, the displaying of graphs and tables is rather verbose as it is made "by hand" using `pandas` or `matplotlib`. I am not an expert in python, but would it be possible to include base plots or summaries for standard outputs of the main functions (such as the overloaded `plot` and `summary` function in `R`.) As the context of analyzing crowdsourced data might involve non-expert in python, I think that might be a useful feature.
- Overall, the arguments of the main functions (`aggregate` and `aggregate-deep`) might deserve a bit of explanation for the reader, has their name (in the current version of the code) is not really explicit (contrary to the arguments of the function `simulate`).



## Minor remarks

- Page 2, item 3 "When the first question is..." should be "When the second question...", I guess;
- Section 2.1: Typo: "In praticular, given ..."
- The `peerannot aggregate ./datasets/cifar10H/ -s GLAD` line code launched the algorithm but after one minute waiting , the first iteration was not completed, is there a problem?"
- Section 4.1.1. In the formula of $CE(u, v)$, shouldn't the role of $u_k$ and $v_k$ be switched, otherwise, this means we would compute, in the formula just above, $\log y_{i, k}^{(j)}$, which is 0 or 1, as far as I understand.
- Section 4.1.2. I think $w_i^{(j)}$ should be $\omega_{i}^{(j)}$.

**Comments On Reproducibility:**

From a reproductible point of view, the authors provide all working codes and all the results and figure depicted in the articles can be reproduced.

**Reproducibility:**

Yes

**Strengths And Weaknesses:**

In my opinion, this work aims at fitting one of the scope of *Computo*, namely

- *Software/tutorial papers to present implementations of stats/ML algorithms or to feature the use of a package/toolbox. For such papers we expect not only the description of an existing implementation but also the study of a concrete use case. If applicable, a comparison to related works and appropriate benchmarking are also expected.* (Source: Computo website).

As a first remark, I would say that the *study of a concrete use case* is a bit light in my opinion, as the two datasets `CIPHAR-H10` and `LabelMe` are clearly toy datasets which are barely described and whose purpose is not discuss.
The authors discuss the difficulty of finding datasets in this domain, but if they could not manage to find other ones, it should be interesting to describe more the one used and their potential interest.
Moreover, it is not discussed whether comparable related works exist for this task, and thus this tool is not compared to any other (but comparable tools might not exist).
From this point of view, I am not in favor of an acceptation in the journal from its current form, even if I let the editors judge whether it is suited for *Computo*.
From a reproductible point of view, the authors provide all working codes and all the results and figure depicted in the articles can be reproduced.

This being said, the article, in my opinion, sometimes lack of clarity in its writing and always oscillates between a description of the package and problematics of aggregation in crowdsourcing.
I think it might be valuable to make all necessary reminders about the methods for crowdsourced datasets (which are necessary and, overall, well done), and then dive into the proposed package, with maybe a bit more precision about arguments of the main functions (see my comments below).

As a major confusion for me during my reading is the fact that the authors emphasize in the introduction that: *"With crowdsourced data the ground truth of a task $x_i$ , denoted $y_i$ is unknown, and there is no single label that can be trusted as in standard supervised learning (even on the train set!)"*, and below that : *"For both `CIFAR-10H` and `LabelMe`, the dataset was originally released in classical supervised learning form (without crowdsourcing). These labels are used as ground truth in evaluations and visualizations. However, we emphasize that crowdsourcing strategies do not rely on the ground truth (only on the workers' answers).*
However, from what I understood, all the analysis made using the package are done using the available ground truth, and thus it is not clear how this package can provide nice insights in the (interesting) setting where ground truth do not exist (for instance, how do we train a classification model, ho do we detect a bad worker, etc...). I think this should be greatly clarified by the authors for further submission (in this journal or another one).

---

> ### Author Response · Authors · 2023-11-23
> **Response part 1**
>
> Thank you for your detailed review. We appreciate your insight and respond below to your concerns:
>
> # Major remarks
>
> - **Dataset, ground truth and concrete use case**: as this is your main concern, let us begin with that point. We want to clarify that we **never** use ground truth labels in our work during aggregation or training, as this is unknown information for practitioners. We have modified some of the pictures and part of the text to state that we only use **an estimation** of the true labels when needed. Yet, ground truth is possibly available for researchers in some datasets, for example for CIFAR-10H and LabelMe two of the most popular publicly available datasets in crowdsourcing classification (idem for the Music dataset, also available in `peerannot`). We agree that this is counter-intuitive, but comes from the way such datasets have been created. In such cases, the ground truth labels are available and can be used a posteriori for performance evaluation of the aggregation or agg-deep steps; see also recent works on these datasets (see [1], [2] and [3] for example from 2021/2022/2023). We also would like to emphasize that our goal, here, is not to create a new crowdsourcing dataset but to provide a toolbox to handle and reproduce existing experiments more easily. `peerannot` `aggregate` and `identify` commands can of course be run on a dataset with unknown ground truth. The `identify` one allowing you to detect bad workers for example is illustrated in Fig. 9 and 10 from section 5. We have also modified the structure of the dataset in Listing. 1 which implicitly required a ground truth for the directory containing the files. We have adapted the code accordingly to avoid such unfortunate dependency and are working on integrating more dataset structures in `peerannot` to fit users needs from different fields of crowdsourcing.
> - **Data format**: the format with -1 and the labels is actually a format proposed by Rodrigues et. al (2018) but not the one we recommand. In fact, depending on how the crowdsourcing was done (AmazonMTurk, Toloka, Appen, your own plateform, etc.) the format widely varies. We are working on a module in `peerannot` allowing to fit multiple plateforms format conversions. However, and this is a remark raised in this paper, the community could benefit on using a format that handles large sparse datasets like the `.json` we proposed, to possibly scale to large project (e.g., on Pl@ntnet data, the number of "workers" and taks could reach millions of units).
> - **Comparison with other toolbox**: We added an appendix to compare our framework with the `crowd-kit` library by Toloka. The main differences are that our `identify` module allows to detect bad workers/spammers with the spammer score by Raykar et. al (2017) or the trace confusion. We also provide the `train` command to directly learn (using Pytorch) from the aggregated labels. From labels aggregated with the $agg$ strategy, this command allows users to load the dataset $(x_i, \hat y_i^{agg})$ and train a neural network specifying the optimizer, batch size and other classical hyperparameters. If a test set is available, predictive performance like test accuracy and calibration scores are computed. Crowd-kit does label aggregation on images classification and segmentation, we chose to focus on classification only. Finaly, we provide in Appendix a performance comparison of different aggregation strategies in term of computational time using the Benchopt [4] library on several datasets.
>
>
> **Citations**:
> - [1]: H. Yang, X. Li and W. Pedrycz, "Learning From Crowds With Contrastive Representation," in IEEE Access, vol. 11, pp. 40182-40191, 2023, doi: 10.1109/ACCESS.2023.3269751.
> - [2]: Zhendong, C., & Hongning, W. (2021). Improve Learning from Crowds via Generative Augmentation. Proceedings of the 27th ACM SIGKDDConference on Knowledge Discovery and Data Mining (KDD’21).
> - [3]: Yang, W., Li, C. & Jiang, L. Learning from crowds with decision trees. Knowl Inf Syst 64, 2123–2140 (2022). https://doi.org/10.1007/s10115-022-01701-9
> - [4]: Moreau, Thomas, et al. "Benchopt: Reproducible, efficient and collaborative optimization benchmarks." Advances in Neural Information Processing Systems 35 (2022): 25404-25421.

---

> ### Author Response · Authors · 2023-11-23
> **Response part 2**
>
> - **Oscillations between the package and problematics of aggregation**: we present the topics our library addresses in Section 1, providing the `peerannot` pipeline in Fig. 1. Then, in Section 2 we deep dive into the core of how to setup/install/store datasets, focuses on our library. And then each section is organized with
>     - first: presenting the problem considered in the crowdsourcing setting (label aggregation, learning from crowdsourced data and identification recalled from the introduction)
>     - second: show how `peerannot` can respond to this problem (with respectively the `aggregate`, `aggregate-deep` and `identify` commands)
>
>     We hope that this structures helps as a handbook for our toolbox where users with one of these classical problems can jump to the needed section, have a description of the possible solutions available in the litterature and then how to use `peerannot` for this problem. **We have now added more details on the code to clarify how to start using** `peerannot`
> - **DS likelihood**: We added the second likelihood expression. Both are in fact useful as the EM algorithm starts by estimating the ground truth, and then plugs this estimation into the formulae computed with the likelihood with known ground truth.
> - **Simulations**: To help with the narration, we created an Appendix with more simulations and only provide some examples in the main paper content.
> - **Confusion matrices in section 4**: In Section 4, and more generally in the whole paper, the ground truths **are only used to compute the performance metrics** in the tables. Confusion matrices are estimated by the different strategies (e.g., DS) without using the ground truths, otherwise it would defeat the purpose of using crowdsourcing, or given for visualization in a simulation setting (Fig. 5) where the ground truth is controlled by the user. For both CrowdLayer and CoNAL, element similar a confusion matrices are considered and represented as tensors to be directly integrated as a layer in a neural network architecture. For CoNAL, there is an additional global one (represented as yet another tensor layer), that is not the average (or any other) aggregation of each worker's element. The parameters are estimated through the optimization steps during the training phase of the neural network (via SGD/Adam). We use the words "estimated labels", when meaningful, to avoid any confusion.
> - **Recommendations with bad workers**: When bad workers are identified, there are multiple ways to handle them:
>     - discard them (if they hurt performance, a simple discard should be enough),
>     - exploit their adversity if they are adversarial and not spammers (if they purposely answer the wrong label in a binary setting, DS can switch back their label relying on confusion matrices) as this could prove beneficial,
>     - use them to test the robustness of your aggregation strategy. Comparing the performance with/without them gives an indication of how well your strategy performs with adversarial/noisy crowd.
> - **Code**: Following Reviewer JvrT's recommendation, we have put most figure-related codes in a `utils.py` file, and remind that the final paper is the html webpage (currently compiled at `https://tanglef.github.io/computo_2023`); the html code unrelated to `peerannot` is collapsed/hidden to ease the readability.
> ### Other remarks
> - Thank you for the typos, they are now corrected (including the CE one and the w in 4.1.2 that is $\omega$)
> - **Computational time**: Computing GLAD on CIFAR-10H takes more than a minute. CIFAR-10H is one of the largest available crowdsourcing datasets with 10.000 tasks and 2570 workers. Furthermore GLAD is known to be slower than DS in general (see the added Appendix for time comparisons on two real datasets).
>
> Please let us know if there are any other concerns.

---

### Author Response · Authors · 2024-01-16
**Reviews and discussion period**

Dear reviewers and action editor,

As the discussion period ended in December, we wished to know if we could do anything else to help the review process.
To summarize the current state of the review:
- we are pleased to say that, reviewer JvrT wrote " I approve publication of the article as it stands".
- we responded to questions from reviewer ZPNA and modified the repository with the given remarks and added more explanations on our library and how it responds to a need of a toolbox in crowdsourcing with comparisons with other libraries in time and features. We showed that our library can handle current simulations as well as real datasets used in recent publications and created appendices to help readability  and focus on our library.

If there are any concerns, please do not hesitate to let us know

---

> ### Comment · Action_Editors · 2024-01-16
> **Pending remarks**
>
> Dear authors,
> I thank you for considering the comments and questions raised by the reviewers. It seems to me that the new version proposed effectively addresses most of the comments.
> However, Reviewer ZPNA has raised some concerns that deserve the authors' attention. Some of them concern minor changes (estimated labels and adding the comment regarding bad workers) but the issue regarding the choice of the dataset is important. It is true that the paper would significantly gain impact if the illustration of the package were indeed on an application of interest. Is there any crowd sourced dataset (as the ones suggested by the referee) that you can use to illustrate all the potential of your work on a more specialized dataset (where the performance of the workers highly depends on specific skills, maybe https://theoryandpractice.citizenscienceassociation.org/articles/10.5334/cstp.556).
> I would be pleased to have your feedback on these last comments.
> Best regards

---

> ### Author Response · Authors · 2024-01-18
> **Crowdsourced bird audio dataset and minor modifications**
>
> Thank your for your feedback.
>
> For the remaining minor remarks, indeed, we discussed how to deal with bad workers in our answer to reviewer ZPNA, we integrated some advice in the main paper too.
> Regarding the datasets, we added the appendix 7.4 with the proposed bird audio's data from Lehikoinen et. al (2023) that is specialized and interesting as it is no longer image classification. Each task was about the comparison of two audio files - hence we only applied classical approaches not deep learning related.
> Simple label aggregation strategies seem to perform well on this data as there are few ambiguous tasks. We display some of the ambiguous audios tasks thanks to our `identify` module. We show that simple identification strategies like the entropy do not reflect well the task's ambiguity level as for a task with two disagreeing votes the entropy is maximal. However, using GLAD's estimated difficulty we recover tasks showing more split decisions from many workers.
> We also show that the workers are, by a vast majority, not spammers and perform quite similarly.
>
> We added this dataset as one of the available datasets in the `peerannot` repository (like `CIFAR-10H` and `LabelMe`) with an installation file (that requires the data to be previously downloaded and extracted from zenodo) so that new users can take advantage of this recently released dataset (see https://github.com/peerannot/peerannot/tree/main/datasets/bird_audio).
>
> We hope that this illustration of our library on a third real dataset, this time related to bird audio files, answers your concerns

---

### Comment · Action_Editors · 2024-03-05

Dear Reviewer,

I appreciate your time and valuable feedback on this manuscript. Your contributions have significantly enhanced the overall quality of the research published in Computo

Computo's general policy regarding the reproducibility and transparency of the peer review process implies that your exchanges with the authors will be made public. You can choose to either reveal your anonymity or preserve it. Could you please indicate your preference regarding this potential unmasking?

Best regards,

---

> ### Comment · Reviewer_JvrT · 2024-03-05
>
> Hi,
>
> I choose to reveal my identity for this review.
>
> Best regards.

---

### Note · Reviewer_ZPNA · 2023-12-13

**Comment:**

I thank the authors for the nice rewriting of the paper.
The structure is now real clear, the distinction between ground truth and estimated labels is overall clarified. It is now clearer and the reading is easier.  Comparisons with other libraries in appendix seems relevant to me.

About this discussion between ground truth and I still have a remark:
The authors says that *"we use the words "estimated labels", when meaningful, to avoid any confusion."*.
I think that it should be used *all the time* when estimated labels are used (and not when *meaningful*, for instance, just to be sure, in section 5.2, in the first bullet, the *"trace of the confusion matrix"* is the trace of the estimated confusion matrix, right?)

As another minor remark, the authors gave me a nice answer for the recommendations for bad workers, but I can't see any trace of it in the paper. I think it would be interesting to add few lines about it.

As major remark that still puzzles me is the interest of the real data sets. I understand that the authors goal is *"not to create a new crowdsourcing dataset"*. However, the journal policy states that for software papers *"the study of a concrete use case"* is expected. I think it's a bit disappointing the focus on such data set that has little interest in any other science (or at least, this interest is not discussed). However, I understand that it might be difficult to find interesting use case (does PlantNet has such annotation issues?). I know that the Surfrider foundation cited in Chagneux and al had an online platform for annotation in the Plastics origins projet. Maybe they have such multiple annotation problem.
In brief, I let to the editor the interpretation of what a *"concrete use case"* is.

To end on a positive remark, though, I am grateful to the authors for the careful (and I am sure, useful) rewriting of the paper.

**Audience:**

Yes

**Claims And Evidence:**

Yes

**Decision Recommendation:**

Leaning Accept

---

### Note · Reviewer_JvrT · 2023-12-17

**Comment:**

Thank you for your replies to my comments.

They address all my points.

I have no further comments to make, and I approve publication of the article as it stands.

**Audience:**

Yes

**Claims And Evidence:**

Yes

**Decision Recommendation:**

Accept

---

### Decision · Action_Editor_dHti · 2024-01-26

**Recommendation:** Accept as is

**Comment:**

Regarding the thorough comments of the two  referees and your modification and  answers  to their comments, I am very pleased to inform  you that your paper is accepted in Computo. I will come back to you as soon as we start the production process, that will certify the reproducibility on the Computo server.

**Audience:**

As the popularity of crowdsourcing annotated data increases, your library  will be  of great interest for the community.

**Claims And Evidence:**

Dear authors,

Your paper provides a new and useful library to handle crowdsourcing annotated images, as well as several illustration of teh practical use of this library.

---

> ### Decision · Editors_In_Chief · 2024-01-26
>
> I approve the AE's decision.

---

> ### Author Response · Authors · 2024-01-29
> **Thank you**
>
> Thank you for the approval
> We updated the pdf as requested in the openreview decision email with the `draft` and `published` badges modified in the github repository.
>
> We remain available for future requests during production process.
> Best regards